# Predictive Coding Approximates Backprop along Arbitrary Computation Graphs

## Abstract

Backpropagation of error (backprop) is a powerful algorithm for training machine learning architectures through end-to-end differentiation. Recently it has been shown that backprop in multilayer-perceptrons (MLPs) can be approximated using predictive coding, a biologically-plausible process theory of cortical computation which relies solely on local and Hebbian updates. The power of backprop, however, lies not in its instantiation in MLPs, but rather in the concept of automatic differentiation which allows for the optimisation of any differentiable program expressed as a computation graph. Here, we demonstrate that predictive coding converges asymptotically (and in practice rapidly) to exact backprop gradients on arbitrary computation graphs using only local learning rules. We apply this result to develop a straightforward strategy to translate core machine learning architectures into their predictive coding equivalents. We construct predictive coding CNNs, RNNs, and the more complex LSTMs, which include a non-layer-like branching internal graph structure and multiplicative interactions. Our models perform equivalently to backprop on challenging machine learning benchmarks, while utilising only local and (mostly) Hebbian plasticity. Our method raises the potential that standard machine learning algorithms could in principle be directly implemented in neural circuitry, and may also contribute to the development of completely distributed neuromorphic architectures.

## 1 Introduction

Deep learning has seen stunning successes in the last decade in computer vision (Krizhevsky et al., 2012; Szegedy et al., 2015), natural language processing and translation (Vaswani et al., 2017; Radford et al., 2019; Kaplan et al., 2020), and computer game playing (Mnih et al., 2015; Silver et al., 2017; Schrittwieser et al., 2019; Vinyals et al., 2019). While there is a great variety of architectures and models, they are all trained by gradient descent using gradients computed by automatic differentiation (AD). The key insight of AD is that it suffices to define a forward model which maps inputs to predictions according to some parameters. Then, using the chain rule of calculus, it is possible, as long as every operation of the forward model is differentiable, to differentiate back through the computation graph of the model so as to compute the sensitivity of every parameter in the model to the error at the output, and thus adjust every single parameter to best minimize the total loss. Early models were typically simple artificial neural networks where the computation graph is simply a composition of matrix multiplications and elementwise nonlinearities, and for which the implementation of automatic differentation has become known as 'backpropagation' (or 'backprop'). However, automatic differentiation allows for substantially more complicated graphs to be differentiated through, up to, and including, arbitrary programs (Griewank et al., 1989; Baydin et al., 2017; Paszke et al., 2017; Revels et al., 2016; Innes et al., 2019; Werbos, 1982; Rumelhart and Zipser, 1985; Linnainmaa, 1970). In recent years this has enabled the differentiation through differential equation solvers (Chen et al., 2018; Tzen and Raginsky, 2019; Rackauckas et al., 2019), physics engines (Degrave et al., 2019; Heiden et al., 2019), raytracers (Pal, 2019), and planning algorithms (Amos and Yarats, 2019; Okada et al., 2017). These advances allow the straightforward training of models which intrinsically embody complex processes and which can encode significantly more prior knowledge and structure about a given problem domain than previously possible.

Modern deep learning has also been closely intertwined with neuroscience (Hassabis et al., 2017; Hawkins and Blakeslee, 2007; Richards et al., 2019). The backpropagation algorithm itself arose

as a technique for training multi-layer perceptrons – simple hierarchical models of neurons inspired by the brain (Werbos, 1982). Despite this origin, and its empirical successes, a consensus has emerged that the brain cannot directly implement backprop, since to do so would require biologically implausible connection rules (Crick, 1989). There are two principal problems. Firstly, backprop in the brain appears to require non-local information (since the activity of any specific neuron affects all subsequent neurons down to the final output neuron). It is difficult to see how this information could be transmitted 'backwards' throughout the brain with the required fidelity without precise connectivity constraints. The second problem – the 'weight transport problem' is that backprop through MLP style networks requires identical forward and backwards weights. In recent years, however, a succession of models have been introduced which claim to implement backprop in MLP-style models using only biologically plausible connectivity schemes, and Hebbian learning rules (Liao et al., 2016; Guerguiev et al., 2017; Sacramento et al., 2018; Bengio and Fischer, 2015; Bengio et al., 2017; Ororbia et al., 2020; Whittington and Bogacz, 2019). Of particular significance is Whittington and Bogacz (2017) who show that predictive coding networks – a type of biologically plausible network which learn through a hierarchical process of prediction error minimization – are mathematically equivalent to backprop in MLP models. In this paper we extend this work, showing that predictive coding can not only approximate backprop in MLPs, but can approximate automatic differentiation along arbitrary computation graphs. This means that in theory there exist potentially biologically plausible algorithms for differentiating through arbitrary programs, utilizing only local connectivity. Moreover, in a class of models which we call parameter-linear, which includes many current machine learning models, the required update rules are Hebbian, raising the possibility that a wide range of current machine learning architectures may be faithfully implemented in the brain, or in neuromorphic hardware.

In this paper we provide two main contributions. (i) We show that predictive coding converges to automatic differentiation across arbitrary computation graphs. (ii) We showcase this result by implementing three core machine learning architectures (CNNs, RNNs, and LSTMs) in a predictive coding framework which utilises only local learning rules and mostly Hebbian plasticity.

## 2 PREDICTIVE CODING ON ARBITRARY COMPUTATION GRAPHS

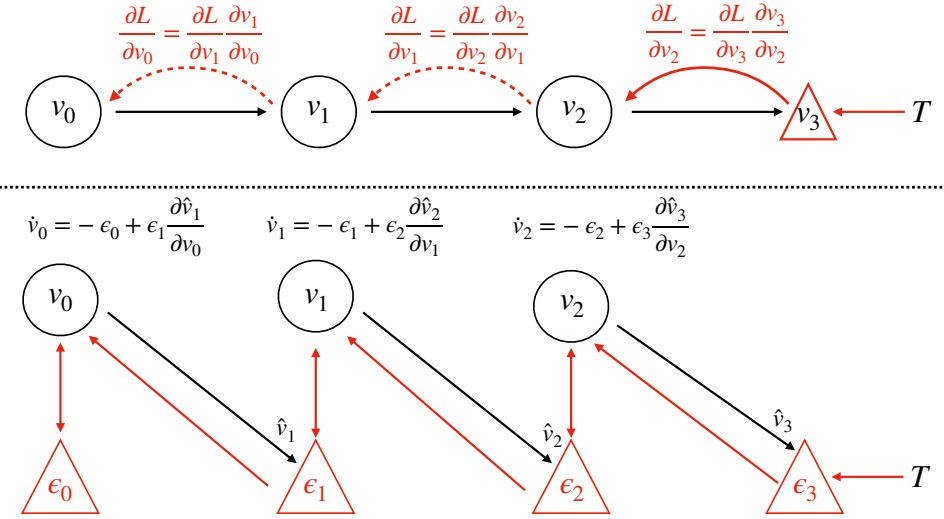

Figure 1: Top: Backpropagation on a chain. Backprop proceeds backwards sequentially and explicitly computes the gradient at each step on the chain. Bottom: Predictive coding on a chain. Predictions, and prediction errors are updated in parallel using only local information.

Predictive coding is an influential theory of cortical function in theoretical and computational neuroscience. Central to the theory is the idea that the core function of the brain is to minimize prediction errors between what is expected to happen and what actually happens. Predictive coding

views the brain as composed of multiple hierarchical layers which predict the activities of the layers below. Unpredicted activity is registered as prediction error which is then transmitted upwards for a higher layer to process. Over time, synaptic connections are adjusted so that the system improves at minimizing prediction error. Predictive coding possesses a wealth of empirical support (Friston, 2003; 2005; Bogacz, 2017; Whittington and Bogacz, 2019) and offers a single mechanism that accounts for diverse perceptual phenomena such as repetition-suppression (Auksztulewicz and Friston, 2016), end-stopping (Rao and Ballard, 1999), bistable perception (Hohwy et al., 2008; Weilnhammer et al., 2017) and illusory motions (Lotter et al., 2016; Watanabe et al., 2018), and even attentional modulation of neural activity (Feldman and Friston, 2010; Kanai et al., 2015). Moreover, the central role of top-down predictions is consistent with the ubiquity, and importance of, top-down diffuse connections between cortical areas. Predictive coding is consistent with many known aspects of neurophysiology, and has been translated into biologically plausible process theories which define candidate cortical microcircuits which can implement the algorithm. (Spratling, 2008; Bastos et al., 2012; Kanai et al., 2015; Shipp, 2016).

In previous work, predictive coding has always been conceptualised as operating on hierarchies of layers (Bogacz, 2017; Whittington and Bogacz, 2017). Here we present a generalized form of predictive coding applied to arbitrary computation graphs. A computation graph $\mathcal{G} = \{\mathbb{E}, \mathbb{V}\}$ is a directed acyclic graph (DAG) which can represent the computational flow of essentially any program or computable function as a composition of elementary functions. Each edge $e_i \in \mathbb{E}$ of the graph corresponds to an intermediate step – the application of an elementary function – while each vertex $v_i \in \mathbb{V}$ is an intermediate variable computed by applying the functions of the edges to the values of their originating vertices. In this paper, $v_i$ denotes the vector of activations within a layer and we denote the set of all vertices as $\{v_i\}$. Effectively, computation flows 'forward' from parent nodes to all their children through the edge functions until the leaf nodes give the final output of the program as a whole (see Figure 1 and 2 for an example). Given a target $T$ and a loss function $L = g(T, v_{out})$, the graph's output can be evaluated and, and if every edge function is differentiable, automatic differentiation can be performed on the computation graph.

Predictive coding can be derived elegantly as a variational inference algorithm under a hierarchical Gaussian generative model (Friston, 2005; Buckley et al., 2017). We extend this approach to arbitrary computation graphs in a supervised setting by defining the inference problem to be solved as that of inferring the vertex value $v_i$ of each node in the graph given fixed start nodes $v_0$ (the data), and end nodes $v_N$ (the targets). We define a generative model which parametrises the value of each vertex given the feedforward prediction of its parents, $p(\{v_i\}) = p(v_0 \dots v_N) = \prod_i^N p(v_i|\mathcal{P}(v_i))$ [1], and a factorised, variational posterior $Q(\{v_i\}|v_0, v_N) = Q(v_1 \dots v_{N-1}|v_0, v_N) = \prod_i^N Q(v_i|\mathcal{P}(v_i), \mathcal{C}(v_i))$, where $\mathcal{P}(v_i)$ denotes the set of parents and $\mathcal{C}(v_i)$ denotes the set of children of a given node $v_i$. From this, we can define a suitable objective functional, the variational free-energy $\mathcal{F}$ (VFE), which acts as an upper bound on the divergence between the true and variational posteriors.

$$\mathcal{F} = KL[(Q(v_1 \dots v_{N-1}|v_0, v_N)\|p(v_0 \dots v_N)] \geq KL[(Q(v_1 \dots v_{N-1})|v_0, v_N)\|p(v_1 \dots v_{N-1}|v_0, v_N)]$$

$$\approx \sum_{i=0}^{N} \epsilon_i^T \epsilon_i$$

$$(1)$$

Under Gaussian assumptions for the generative model $p(\{v_i\}) = \prod_i^N \mathcal{N}(v_i; \hat{v}_i, \Sigma_i)$, and the variational posterior $Q(\{v_i\}) = \prod_i^N \mathcal{N}(v_i)$, where the 'predictions' $\hat{v}_i = f(\mathcal{P}(v_i); \theta_i)$ are defined as the feedforward value of the vertex produced by running the graph forward, and all the precisions, or inverse variances, $\Sigma_i^{-1}$ are fixed at the identity, we can write $\mathcal{F}$ as simply a sum of prediction errors (see Appendix D or (Friston, 2003; Bogacz, 2017; Buckley et al., 2017) for full derivations), with the prediction errors defined as $\epsilon_i = v_i - \hat{v}_i$. These prediction errors play a core role in the framework and, in the biological process theories (Friston, 2005; Bastos et al., 2012), are generally considered to be represented by a distinct population of 'error units'. Since $\mathcal{F}$ is an upper bound on the divergence between true and approximate posteriors, by minimizing $\mathcal{F}$, we reduce this divergence, thus improving the quality of the variational posterior and approximating exact Bayesian inference. Predictive coding minimizes $\mathcal{F}$ by employing the Cauchy method of steepest descent to set the

---

[1]This includes the prior $p(v_0)$, which simply has no parents.

dynamics of the vertex variables $v_i$ as a gradient descent directly on $\mathcal{F}$ (Bogacz, 2017).

$$\frac{dv_i}{dt} = \frac{\partial \mathcal{F}}{\partial v_i} = \epsilon_i - \sum_{j \in \mathcal{C}(v_i)} \epsilon_j \frac{\partial \hat{v}_j}{\partial v_i} \tag{2}$$

The dynamics of the parameters of the edge functions $\theta$ such that $\hat{v}_i = f(\mathcal{P}(v_i); \theta)$, can also be derived as a gradient descent on $\mathcal{F}$. Importantly these dynamics require only information (the current vertex value, prediction error, and prediction errors of child vertices) locally available at the vertex.

$$\frac{d\theta_i}{dt} = \frac{\partial \mathcal{F}}{\partial \theta_i} = \epsilon_i \frac{\partial \hat{v}_i}{\partial \theta_i} \tag{3}$$

To run generalized predictive coding in practice on a given computation graph $\mathcal{G} = \{\mathbb{E}, \mathbb{V}\}$, we augment the graph with error units $\epsilon \in \mathcal{E}$ to obtain an augumented computation graph $\tilde{\mathcal{G}} = \{\mathbb{E}, \mathbb{V}, \mathcal{E}\}$. The predictive coding algorithm then operates in two phases – a feedforward sweep and a backwards iteration phase. In the feedforward sweep, the augmented computation graph is run forward to obtain the set of predictions $\{\hat{v}_i\}$, and prediction errors $\{\epsilon_i\} = \{v_i - \hat{v}_i\}$ for every vertex. Following Whittington and Bogacz (2017), to achieve exact equivalence with the backprop gradients computed on the original computation graph, we initialize $v_i = \hat{v}_i$ in the initial feedforward sweep so that the output error computed by the predictive coding network and the original graph are identical.

In the backwards iteration phase, the vertex activities $\{v_i\}$ and prediction errors $\{\epsilon_i\}$ are updated with Equation 2 for all vertices in parallel until the vertex values converge to a minimum of $\mathcal{F}$. After convergence the parameters are updated according to Equation 3. Note we also assume, following Whittington and Bogacz (2017), that the predictions at each layer are fixed at the values assigned during the feedforward pass throughout the optimisation of the $v$s. We call this the *fixed-prediction assumption*. In effect, by removing the coupling between the vertex activities of the parents and the prediction at the child, this assumption separates the global optimisation problem into a local one for each vertex. We implement these dynamics with a simple forward Euler integration scheme so that the update rule for the vertices became $v_i^{t+1} \leftarrow v_i^t - \eta \frac{d\mathcal{F}}{dv_i^t}$ where $\eta$ is the step-size parameter. Importantly, if the edge function linearly combines the activities and the parameters followed by an elementwise nonlinearity – a condition which we call 'parameter-linear' – then both the update rule for the vertices (Equation 2) and the parameters (Equation 3) become Hebbian. Specifically, the update rules for the vertices and weights become $\frac{dv_i}{dt} = \epsilon_i - \sum_j \epsilon_j f'(\theta_j \hat{v}_j) \theta_j^T$ and $\frac{d\theta_i}{dt} = \epsilon_i f'(\theta_i \hat{v}_i) \hat{v}_i^T$, respectively.

## 2.1 APPROXIMATION TO BACKPROP

Here we show that at the equilibrium of the dynamics, the prediction errors $\epsilon_i^*$ converge to the correct backpropagated gradients $\frac{\partial L}{\partial v_i}$, and consequently the parameter updates (Equation 3) become precisely those of a backprop trained network. Standard backprop works by computing the gradient of a vertex as the sum of the gradients of the child vertices. Beginning with the gradient of the output vertex $\frac{\partial L}{\partial v_L}$, it recursively computes the gradients of vertices deeper in the graph by the chain rule:

$$\frac{\partial L}{\partial v_i} = \sum_{j=\mathcal{C}(v_i)} \frac{\partial L}{\partial v_j} \frac{\partial v_j}{\partial v_i} \tag{4}$$

In comparison, in our predictive coding framework, at the equilibrium point ($\frac{dv_i}{dt} = 0$) the prediction errors $\epsilon_i^*$ become,

$$\epsilon_i^* = \sum_{j \in \mathcal{C}(v_i)} \epsilon_j^* \frac{\partial \hat{v}_i}{\partial v_j} \tag{5}$$

Importantly, this means that the equilibrium value of the prediction error at a given vertex (Equation 5) satisfies the same recursive structure as the chain rule of backprop (Equation 4). Since this relationship is recursive, all that is needed for the prediction errors throughout the graph to converge to the backpropagated derivatives is for the prediction errors at the final layer to be equal to the output gradient: $\epsilon_L^* = \frac{\partial L}{\partial \hat{v}_L}$. To see this explicitly, consider a mean-squared-error loss function [2]. at the

---

[2]While the mean-squared-error loss function fits most nicely with the Gaussian generative model, other loss functions can be used in practice. If the loss function can be represented as a log probability distribution, then the generative model can be amended to simply set the output distribution to that distribution. If not, then there is no fully consistent generative model (although all nodes except the output remain Gaussian), but the algorithm will still work in practice. See (Figure 6) in Appendix A for results for CNNs trained with a crossentropy loss.

---

**Algorithm 1:** Generalized Predictive Coding

---

**Data:** Dataset $\mathcal{D} = \{\mathbf{X}, \mathbf{L}\}$, Augmented Computation Graph $\tilde{\mathcal{G}} = \{\mathbb{E}, \mathbb{V}, \mathcal{E}\}$, inference learning
rate $\eta_v$, weight learning rate $\eta_\theta$

**begin**

```
/* For each minibatch in the dataset                        */
for (x, L) ∈ D do
    /* Fix start of graph to inputs                         */
    v̂₀ ← x
    /* Forward pass to compute predictions                  */
    for v̂ᵢ ∈ V do
      └ v̂ᵢ ← f({P(v̂ᵢ); θ})
    /* Compute output error                                 */
    ε_L ← L − v̂_L
    /* Begin backwards iteration phase of the descent on the
       free energy                                          */
    while not converged do
        for (vᵢ, εᵢ) ∈ G̃ do
            /* Compute prediction errors                    */
            εᵢ ← vᵢ − v̂ᵢ
            /* Update the vertex values                     */
          └ vᵢ^{t+1} ← vᵢ^t + η_v (dF/dvᵢ^t)

    /* Update weights at equilibrium                        */
    for θᵢ ∈ E do
      └ θᵢ^{t+1} ← θᵢ^t + η_θ (dF/dθᵢ^t)
```

---

output layer $L = \frac{1}{2}(T - \hat{v}_L)^2$ with T as a vector of targets, and defining $\epsilon_L = T - \hat{v}_L$. We then consider the equilibrium value of the prediction error unit at a penultimate vertex $\epsilon_{L-1}$. By Equation 5, we can see that at equilibrium,

$$\epsilon_{L-1}^* = \epsilon_L^* \frac{\partial \hat{v}_L}{\partial v_{L-1}} = (T - \hat{v}_L^*) \frac{\partial \hat{v}_L}{\partial v_{L-1}}$$

since, $(T - \hat{v}_L) = \frac{\partial L}{\partial \hat{v}_L}$, we can then write,

$$\epsilon_{L-1}^* = \frac{\partial L}{\partial \hat{v}_L} \frac{\partial \hat{v}_L}{\partial v_{L-1}} = \frac{\partial L}{\partial v_{L-1}} \tag{6}$$

Thus the prediction errors of the penultimate nodes converge to the correct backpropagated gradient. Furthermore, recursing through the graph from children to parents allows the correct gradients to be computed[3]. Thus, by induction, we have shown that the fixed points of the prediction errors of the global optimization correspond exactly to the backpropagated gradients. Intuitively, if we imagine the computation-graph as a chain and the error as 'tension' in the chain, backprop loads all the tension at the end (the output) and then systematically propagates it backwards. Predictive coding, however, spreads the tension throughout the entire chain until it reaches an equilibrium where the amount of tension at each link is precisely the backpropagated gradient. The full algorithm for training the predictive coding network is explicitly set out in Algorithm 1. Inference is just a forward pass through the network, and is identical to the corresponding ANN.

---

[3]Some subtlety is needed here since $v_{L-1}$ may have many children which each contribute to the loss. However, these different paths sum together at the node $v_{L-1}$, thus propagating the correct gradient backwards.

By a similar argument, it is apparent that the dynamics of the parameters $\theta_i$ as a gradient descent on $\mathcal{F}$ also exactly match the backpropagated parameter gradients.

$$
\begin{aligned}
\frac{d\theta_i}{dt} = \frac{d\mathcal{F}}{d\theta_i} &= \epsilon_i^* \frac{d\epsilon_i^*}{d\theta_i} \\
&= \frac{dL}{d\hat{v}_i} \frac{d\hat{v}_i}{d\theta_i} = \frac{dL}{d\theta_i}
\end{aligned}
\tag{7}
$$

Which follows from the fact that $\epsilon_i^* = \frac{dL}{d\hat{v}_i}$ and that $\frac{d\epsilon_i^*}{d\theta} = \frac{d\hat{v}_i}{d\theta_i}$.

## 3    RELATED WORK

A number of recent works have tried to provide biologically plausible approximations to backprop. The requirement of symmetry between the forwards and backwards weights has been questioned by Lillicrap et al. (2016) who show that random fixed feedback weights suffice for effective learning. Recent additional work has shown that learning the backwards weights also helps (Amit, 2019; Akrout et al., 2019). Several schemes have also been proposed to approximate backprop using only local learning rules and/or Hebbian connectivity. These include target-prop (Lee et al., 2015) which approximate the backward gradients with trained inverse functions, but which fails to asymptotically compute the exact backprop gradients, and contrastive Hebbian (Seung, 2003; Scellier and Bengio, 2017; Scellier et al., 2018) approaches which do exactly approximate backprop, but which require two separate learning phases and the storing of information across successive phases. There are also dendritic error theories (Guerguiev et al., 2017; Sacramento et al., 2018) which are computationally similar to predictive coding (Whittington and Bogacz, 2019; Lillicrap et al., 2020). Whittington and Bogacz (2017) showed that predictive coding can approximate backprop in MLP models, and demonstrated comparable performance on MNIST. We advance upon this work by extending the proof to arbitrary computation graphs, enabling the design of predictive coding variants of a range of standard machine learning architectures, which we show perform comparably to backprop on considerably more difficult tasks than MNIST. Our algorithm evinces asymptotic (and in practice rapid) convergence to the exact backprop gradients, does not require separate learning phases, and utilises only local information and largely Hebbian plasticity.

## 4    RESULTS

### 4.1    NUMERICAL RESULTS

To demonstrate the correctness of our derivation and empirical convergence to the true gradients, we present a numerical test in the simple scalar case, where we use predictive coding to derive the gradients of an arbitrary, highly nonlinear test function $v_L = \tan(\sqrt{\theta v_0}) + \sin(v_0^2)$ where $\theta$ is an arbitrary parameter. For our tests, we set $v_0$ to 5 and $\theta$ to 2. The computation graph for this function is presented in Figure 2. Although simple, this is a good test of predictive coding because the function is highly nonlinear, and its computation graph does not follow a simple layer structure but includes some branching. An arbitrary target of $T = 3$ was set at the output and the gradient of the loss $L = (v_L - T)^2$ with respect to the input $v_0$ was computed by predictive coding. We show (Figure 2) that the predictive coding optimisation rapidly converges to the exact numerical gradients computed by automatic differentiation, and that moreover this optimization is very robust and can handle even exceptionally high learning rates (up to 0.5) without divergence.

In summary, we have shown and numerically verified that at the equilibrium point of the global free-energy $\mathcal{F}$ on an arbitrary computation graph, the error units exactly equal the backpropagated gradients, and that this descent requires only local connectivity, does not require a separate phases or a sequential backwards sweep, and in the case of parameter-linear functions, requires only Hebbian plasticity. Our results provide a straightforward recipe for the direct implementation of predictive coding algorithms to approximate certain computation graphs, such as those found in common machine learning algorithms, in a potentially biologically plausible manner. Next, we showcase this capability by developing predictive coding variants of core machine learning architectures - convolutional neural networks (CNNs) recurrent neural networks (RNNs) and LSTMs (Hochreiter and Schmidhuber, 1997), and show performance comparable with backprop on tasks substantially more challenging than MNIST.

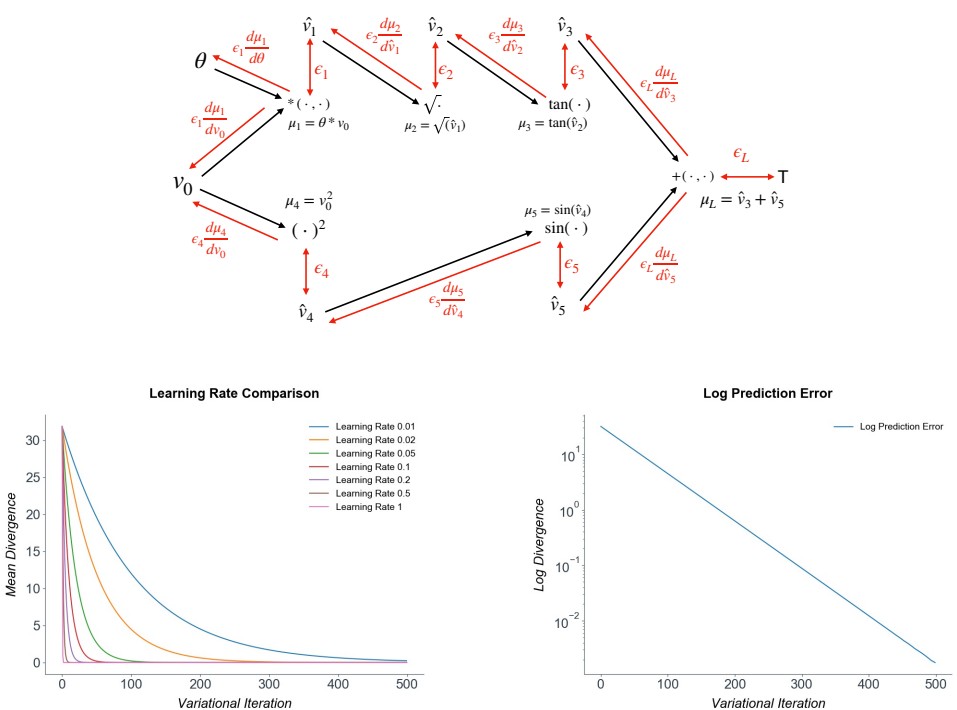

Figure 2: Top: The computation graph of the nonlinear test function $v_L = \tan(\sqrt{\theta v_0}) + \sin(v_0^2)$. Bottom: graphs of the log mean divergence from the true gradient and the divergence for different learning rates. Convergence to the exact gradients is exponential and robust to high learning rates.

## 4.2 PREDICTIVE CODING CNN, RNN, AND LSTM

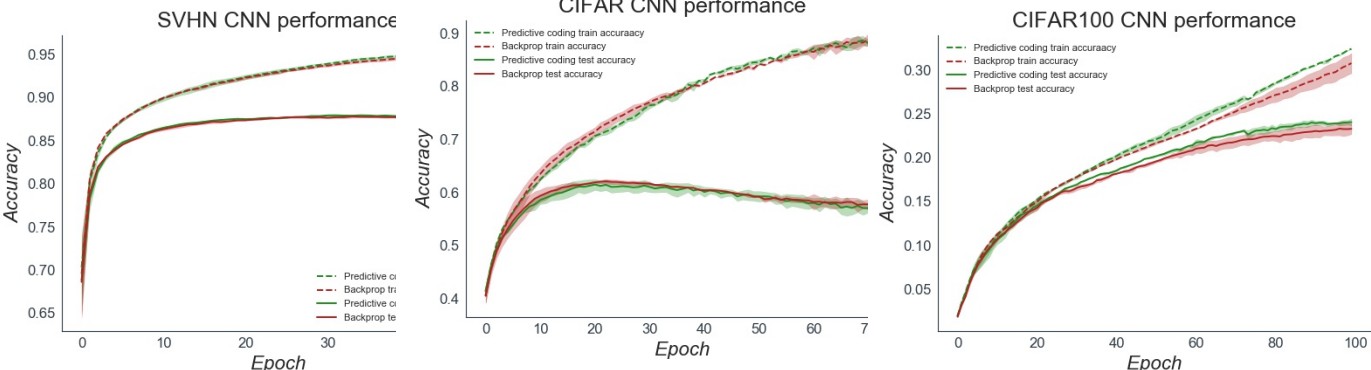

Figure 3: Training and test accuracy plots for the Predictive Coding and Backprop CNN on SVHN,CIFAR10, and CIFAR10 dataest over 5 seeds. Performance is largely indistinguishable. Due to the need to iterate the $v$s until convergence, the predictive coding network had roughly a 100x greater computational cost than the backprop network.

First, we constructed predictive coding CNN models (see Appendix B for full implementation details). In the predictive coding CNN, each filter kernel was augmented with 'error maps' which measured the difference between the forward convolutional predictions and the backwards messages. Our CNN was composed of a convolutional layer, followed by a max-pooling layer, then two further convolutional layers followed by 3 fully-connected layers. We compared our predictive coding CNN to a backprop-trained CNN with the exact same architecture and hyperparameters. We tested our models on three image classification datasets significantly more challenging than MNIST – SVHN, CIFAR10, and CIFAR100. SVHN is a digit recognition task like MNIST, but has more naturalistic backgrounds, is in colour with continuously varying inputs and contains distractor digits. CIFAR10 and CIFAR100 are large image datasets composed of RGB 32x32 images. CIFAR10 has

10 classes of image, while CIFAR100 is substantially more challenging with 100 possible classes. In general (Figure 3), performance was identical between the predictive coding and backprop CNNs and comparable to the standard performance of basic CNN models on these datasets, Moreover, the predictive coding gradient remained close to the true numerical gradient throughout training.

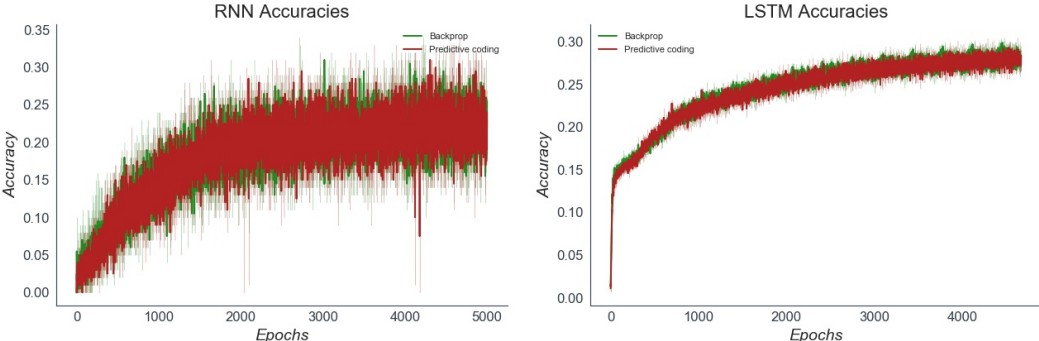

Figure 4: Test accuracy plots for the Predictive Coding and Backprop RNN and LSTM on their respective tasks, averaged over 5 seeds. Performance is again indistinguishable from backprop.

We also constructed predictive coding RNN and LSTM models, thus demonstrating the ability of predictive coding to scale to non-parameter-linear, branching, computation graphs. The RNN was trained on a character-level name classification task, while the LSTM was trained on a next-character prediction task on the full works of Shakespeare. Full implementation details can be found in Appendices B and C. LSTMs and RNNs are recurrent networks which are trained through backpropagation through time (BPTT). BPTT simply unrolls the network through time and backpropagates through the unrolled graph. Analogously we trained the predictive coding RNN and LSTM by applying predictive coding to the unrolled computation graph. The depth of the unrolled graph depends heavily on the sequence length, and in our tasks using a sequence length of 100 we still found that predictive coding evinced rapid convergence to the correct numerical gradient, and that the performance was approximately identical to the equivalent backprop-trained networks (Figure 3), thus showing that the algorithm is scalable even to very deep computation graphs.

## 5 DISCUSSION

We have shown that predictive coding provides a local and potentially biologically plausible approximation to backprop on arbitrary, deep, and branching computation graphs. Moreover, convergence to the exact backprop gradients is rapid and robust, even in extremely deep graphs such as the unrolled LSTM. Our algorithm is fully parallelizable, does not require separate phases, and can produce equivalent performance to backprop in core machine-learning architectures. These results broaden the horizon of local approximations to backprop by demonstrating that they can be implemented on arbitrary computation graphs, not only simple MLP architectures. Our work prescribes a straight-forward recipe for backpropagating through any computation graph with predictive coding using only local learning rules. In the future, this process could potentially be made fully automatic and translated onto neuromorphic hardware. Our results also raise the possibility that the brain may implement machine-learning type architectures much more directly than often considered. Many lines of work suggest a close correspondence between the representations and activations of CNNs and activity in higher visual areas (Yamins et al., 2014; Tacchetti et al., 2017; Eickenberg et al., 2017; Khaligh-Razavi and Kriegeskorte, 2014; Lindsay, 2020), for instance, and this similarity may be found to extend to other machine learning architectures.

It is important to note that predictive coding, as advanced here, still retains some biologically implausible features. Although using only local and Hebbian updates, the predictive coding algorithm still requires identical forward and backwards weights, as well as mandating a very precise one-to-one connectivity structure between value neurons $v_i$ and error neurons $\epsilon_i$. However, recent work (Millidge et al., 2020) has begun to show that these implausibilities can be relaxed using learnable backwards weights instead of requiring weight symmetry, and allowing for learnable dense connectivity between value and error neurons, without harm to performance in simple MLP settings. An additional limitation to the biological plausibility of our method is the fixed-prediction assumption, which requires that the feedforward pass values be somehow stored during the backwards

iteration phase. In biological neurons this could potentially be implemented by utilizing synaptic mechanisms for maintaining information over short periods, such as eligibility traces, or alternatively through synchronised phase locking (Buzsaki, 2006). Alternatively, it is important to note that this fixed-prediction assumption is only required for *exact* convergence to backprop, and predictive coding networks have been shown to be able to attain strong discriminative classification performance without it (Whittington and Bogacz, 2017).

Although we have implemented three core machine learning architectures as predictive coding networks, we have nevertheless focused on relatively small and straightforward networks and thus both our backprop and predictive coding networks perform below the state of the art on the presented tasks. This is primarily because our focus was on demonstrating the theoretical convergence between the two algorithms. Nevertheless, we believe that due to the generality of our theoretical results, 'scaling up' the existing architectures to implement performance-matched predictive coding versions of more advanced machine learning architectures such as resnets (He et al., 2016), GANs (Goodfellow et al., 2014), and transformers (Vaswani et al., 2017) should be relatively straightforward.

In terms of computational cost, one inference iteration in the predictive coding network is about as costly as a backprop backwards pass. Thus, due to using 100-200 iterations for full convergence, our algorithm is substantially more expensive than backprop which limits the scalability of our method. However, this serial cost is misleading when talking about highly parallel neural architectures. In the brain, neurons cannot wait for a sequential forward and backward sweep. By phrasing our algorithm as a global descent, our algorithm is fully parallel across layers. There is no waiting and no phases to be coordinated. Each neuron need only respond to its local driving inputs and downwards error signals. We believe that this local and parallelizable property of our algorithm may engender the possibility of substantially more efficient implementations on neuromorphic hardware (Furber et al., 2014; Merolla et al., 2014; Davies et al., 2018), which may ameliorate much of the computational overhead compared to backprop. Future work could also examine whether our method is more capable than backprop of handling the continuously varying inputs the brain is presented with in practice, rather than the artificial paradigm of being presented with a series of *i.i.d.* datapoints.

Our work also reveals a close connection between backprop and inference. Namely, the recursive computation of gradients is effectively a by-product of a variational-inference algorithm which infers the values of the vertices of the computation graph under a hierarchical Gaussian generative model. While the deep connections between stochastic gradient descent and inference in terms of Kalman filtering (Ruck et al., 1992; Ollivier, 2019) or MCMC sampling methods (Chen et al., 2014; Mandt et al., 2017) is known, the relation between recursive gradient computation itself and variational inference is underexplored except in the case of a single layer (Amari, 1995). Our method can provide a principled generalisation of backprop through the inverse-variance $\Sigma^{-1}$ parameters of the Gaussian generative model. These parameters weight the relative contribution of different factors to the overall gradient by their uncertainty, thus naturally handling the case of backprop with differentially noisy inputs. Moreover, the $\Sigma^{-1}$ parameters can be learnt as a gradient descent on $\mathcal{F}$: $\frac{d\Sigma_i}{dt} = -\frac{d\mathcal{F}}{d\Sigma_i} = -\Sigma_i^{-1}\epsilon_i\epsilon_i^T\Sigma_i^{-1} - \Sigma_i^{-1}$. This specific generalisation is afforded by the Gaussian form of the generative model, however, and other generative models may yield novel optimisation algorithms able to quantify and handle uncertainties throughout the entire computational graph.

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

## APPENDIX A: PREDICTIVE CODING CNN IMPLEMENTATION DETAILS

The key concept in a CNN is that of an image convolution, where a small weight matrix is 'slid' (or convolved) across an image to produce an output image. Each patch of the output image only depends on a relatively small patch of the input image. Moreover, the weights of the filter stay the same during the convolution, so each pixel of the output image is generated using the same weights. The weight sharing implicit in the convolution operation enforces translational invariance, since different image patches are all processed with the same weights.

The forward equations of a convolutional layer for a specific output pixel

$$v_{i,j} = \sum_{k=i-f}^{k=i+f} \sum_{l=j-f}^{l=j+f} \theta_{k,l} x_{i+k,j+l}$$

Where $v_{i,j}$ is the $(i,j)$th element of the output, $x_{i,j}$ is the element of the input image and $\theta_{k,l}$ is an weight element of a feature map. To setup a predictive coding CNN, we augment each intermediate $x_i$ and $v_i$ with error units $\epsilon_i$ of the same dimension as the output of the convolutional layer.

Predictions $\hat{v}$ are projected forward using the forward equations. Prediction errors also need to be transmitted backwards for the architecture to work. To achieve this we must have that prediction errors are transmitted upwards by a 'backwards convolution'. We thus define the backwards prediction errors $\hat{\epsilon}_j$ as follows:

$$\hat{\epsilon}_{i,j} = \sum_{k=i-f}^{i+f} \sum_{l=j-f}^{j+f} \theta_{j,i} \tilde{\epsilon}_{i,j}$$

Where $\tilde{\epsilon}$ is an error map zero-padded to ensure the correct convolutional output size. Inference in the predictive coding network then proceeds by updating the intermediate values of each layer as follows:

$$\frac{dv_l}{dt} = \epsilon_l - \hat{\epsilon}_{l+1}$$

Since the CNN is also parameter-linear, weights can be updated using the simple Hebbian rule of the multiplication of the pre and post synaptic potentials.

$$\frac{d\theta_l}{dt} = \sum_{i,j} \epsilon_{l_{i,j}} v_{l-1}{}_{i,j}^T$$

There is an additional biological implausibility here due to the weight sharing of the CNN. Since the same weights are copied for each position on the image, the weight updates have contributions from all aspects of the image simultaneously which violates the locality condition. A simple fix for this, which makes the network scheme plausible is to simply give each position on the image a filter with separate weights, thus removing the weight sharing implicit in the CNN. In effect this gives each patch of pixels a local receptive field with its own set of weights. The performance and scalability

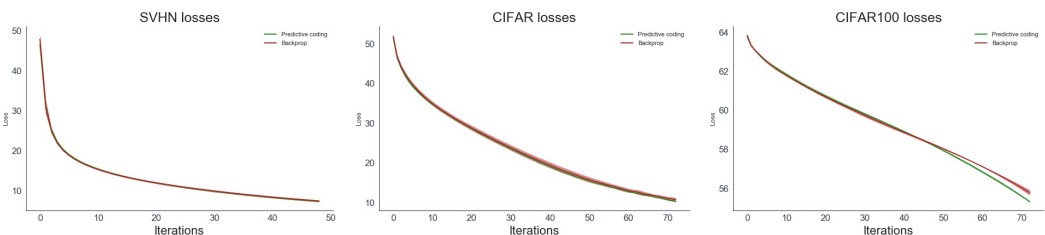

Figure 5: Training loss plots for the Predictive Coding and Backprop CNN on SVHN,CIFAR10, and CIFAR10 dataset over 5 seeds.

of such a locally connected predictive coding architecture would be an interesting avenue for future work, as this architecture has substantial homologies with the structure of the visual cortex.

In our experiments we used a relatively simple CNN architecture consisting of one convolutional layer of kernel size 5, and a filter bank of 6 filters. This was followed by a max-pooling layer with a (2,2) kernel and a further convolutional layer with a (5,5) kernel and filter bank of 16 filters. This was then followed by three fully connected layers of 200, 150, and 10 (or 100 for CIFAR100) output units. Each convolutional and fully connected layer used the relu activation function, except the output layer which was linear. Although this architecture is far smaller than state of the art for convolutional networks, the primary point of our paper was to demonstrate the equivalence of predictive coding and backprop. Further work could investigate scaling up predictive coding to more state-of-the-art architectures.

Our datasets consisted of 32x32 RGB images. We normalised the values of all pixels of each image to lie between 0 and 1, but otherwise performed no other image preprocessing. We did not use data augmentation of any kind. We set the weight learning rate for the predictive coding and backprop networks 0.0001. A minibatch size of 64 was used. These parameters were chosen without any detailed hyperparameter search and so are likely suboptimal. The magnitude of the gradient updates was clamped to lie between -50 and 50 in all of our models. This was done to prevent divergences, as occasionally occurred in the LSTM networks, likely due to exploding gradients.

The predictive coding scheme converged to the exact backprop gradients very precisely within 100 inference iterations using an inference learning rate of 0.1. This gives the predictive coding CNN approximately a 100x computational overhead compared to backprop. The divergence between the true and approximate gradients remained approximately constant throughout training, as shown by Figure 5, which shows the mean divergence for each layer of the CNN over the course of an example training run on the CIFAR10 dataset. The training loss of the predictive coding and backprop networks for SVHN, CIFAR10 and CIFAR100 are presented in Figure 4.

While the experiments in the main paper all used the mean-squared-error loss function, it is also possible to use alternative loss functions. In Figure 6, we show performance of the CNN on CIFAR and SVHN datasets is also very close to backprop when trained with a multi-class cross-entropy loss $L = \sum_i T_i \ln v_{Li}$. In this case the output layer used a softmax function as its nonlinearity, to ensure that the logits passed to the cross-entropy loss were valid probabilities. The cross-entropy loss is also straightforward to fit into the predictive coding framework since the gradient with respect to the pre-activations of the output is also just the negative prediction error $\frac{\delta L}{\partial v_L} = T - v_L$, although the softmax function itself may be challenging to implement neurally since it is non-local as its' normalisation coefficient requires of the exponentiated activities of all neurons in a layer. Nevertheless, this demonstrates that predictive coding can approximate backprop for any given loss function, not simply mean-square-error.

## APPENDIX B: PREDICTIVE CODING RNN

The computation graph on RNNs is relatively straightforward. We consider only a single layer RNN here although the architecture can be straightforwardly extended to hierarchically stacked RNNs. An RNN is similar to a feedforward network except that it possesses an additional hidden state $h$ which is maintained and updated over time as a function of both the current input $x$ and the previous hidden

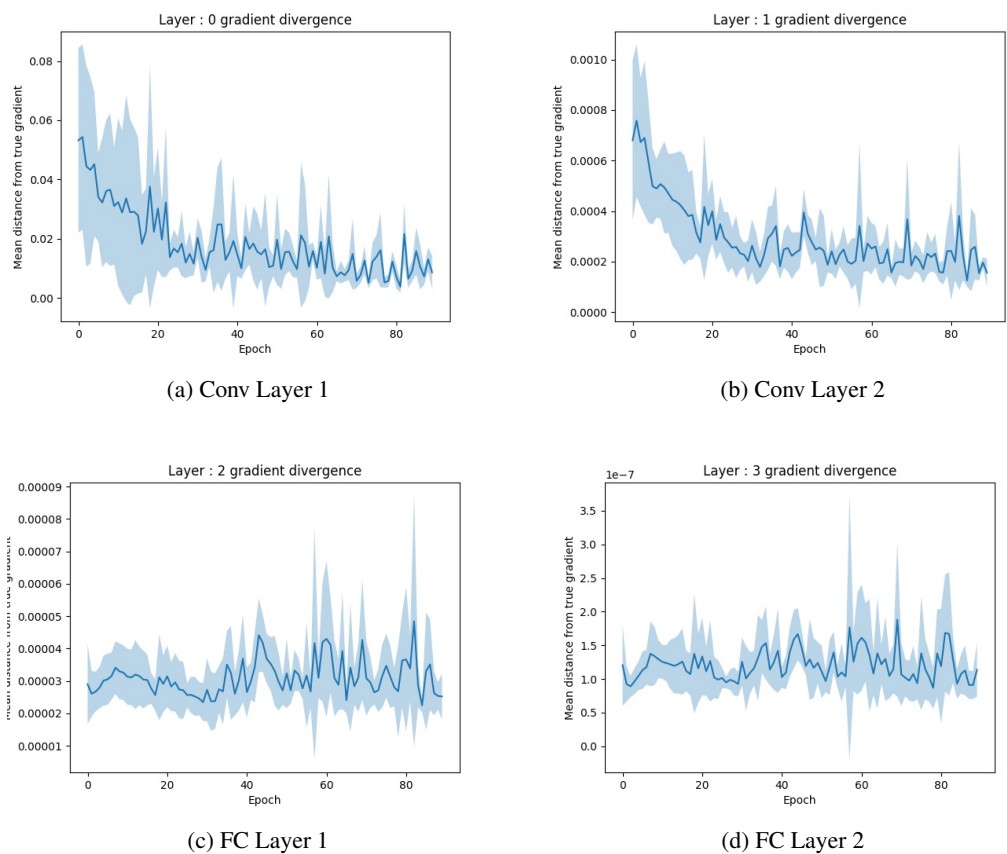

(a) Conv Layer 1

(b) Conv Layer 2

(c) FC Layer 1

(d) FC Layer 2

Mean divergence between the true numerical and predictive coding backprops over the course of training. In general, the divergence appeared to follow a largely random walk pattern, and was generally neglible. Importantly, the divergence did not grow over time throughout training, implying that errors from slightly incorrect gradients did not appear to compound.

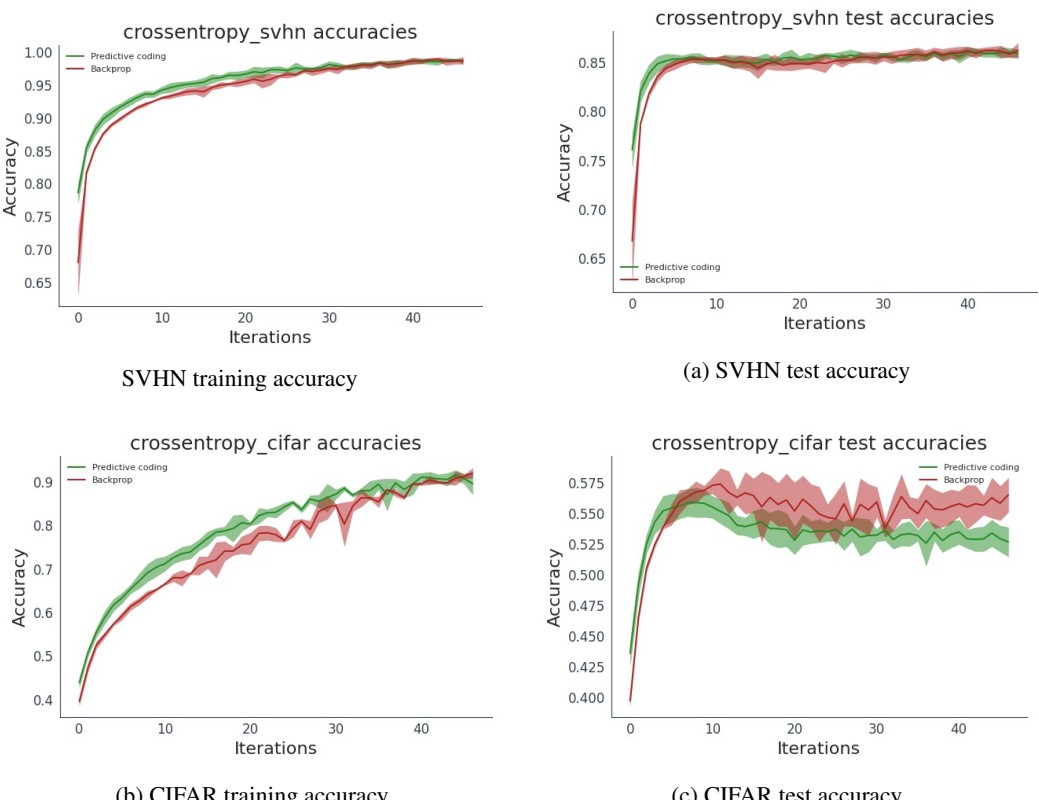

SVHN training accuracy

(a) SVHN test accuracy

(b) CIFAR training accuracy

(c) CIFAR test accuracy

Training and test accuracies of the CNN network on the SVHN and CIFAR datasets using the cross-entropy loss. As can be seen performance remains very close to backprop, thus demonstrating that our predictive coding algorithm can be used with different loss functions, not just mean-squared-error.

state. The output of the network $y$ is a function of $h$. By considering the RNN at a single timestep we obtain the following equations.

$$h_t = f(\theta_h h_{t-1} + \theta_x x_t) \tag{8}$$
$$y_t = g(\theta_y h_t)$$

Where f and g are elementwise nonlinear activation functions. And $\theta_h, \theta_x, \theta_y$ are weight matrices for each specific input. To predict a sequence the RNN simply rolls forward the above equations to generate new predictions and hidden states at each timestep.

RNNs are typically trained through an algorithm called backpropagation through time (BPTT) which essentially just unrolls the RNN into a single feedforward computation graph and then performs backpropagation through this unrolled graph. To train the RNN using predictive coding we take the same approach and simply apply predictive coding to the unrolled graph.

It is important to note that this is an additional aspect of biological implausibility that we do not address in this paper. BPTT requires updates to proceed backwards through time from the end of the sequence to the beginning. Ignoring any biological implausibility with the rules themselves, this updating sequence is clearly not biologically plausible as naively it requires maintaining the entire sequence of predictions and prediction errors perfectly in memory until the end of the sequence, and waiting until the sequence ends before making any updates. There is a small literature on trying to produce biologically plausible, or forward-looking approximations to BPTT which does not require updates to be propagated back through time (Williams and Zipser, 1989; Lillicrap and Santoro, 2019; Steil, 2004; Ollivier et al., 2015; Tallec and Ollivier, 2017). While this is a fascinating area, we do not address it in this paper. We are solely concerned with the fact that predictive coding approximates backpropagation on feedforward computation graphs for which the unrolled RNN graph is a sufficient substrate.

To learn a predictive coding RNN, we first augment each of the variables $h_t$ and $y_t$ of the original graph with additional error units $\epsilon_{h_t}$ and $\epsilon_{y_t}$. Predictions $\hat{y}_t, \hat{h}_t$ are generated according to the feedforward rules (16). A sequence of true labels $\{T_1...T_T\}$ is then presented to the network, and then inference proceeds by recursively applying the following rules backwards through time until convergence.

$$\epsilon_{y_t} = L - \hat{y}_t$$
$$\epsilon_{h_t} = h_t - \hat{h}_t$$
$$\frac{dh_t}{dt} = \epsilon_{h_t} - \epsilon_{y_t}\theta_y^T - \epsilon_{h_{t+1}}\theta_h^T$$

Upon convergence the weights are updated according to the following rules.

$$\frac{d\theta_y}{dt} = \sum_{t=0}^{T} \epsilon_{y_t}\frac{\partial g(\theta_y h_t)}{\partial\theta_y}h_t^T$$

$$\frac{d\theta_x}{dt} = \sum_{t=0}^{T} \epsilon_{h_t}\frac{\partial f(\theta_h h_{t-1} + \theta_x x_t)}{\partial\theta_x}x_t^T$$

$$\frac{d\theta_h}{dt} = \sum_{t=0}^{T} \epsilon_{h_t}\frac{\partial f(\theta_h h_{t-1} + \theta_x x_t)}{\partial\theta_h}h_{t+1}^T$$

Since the RNN feedforward updates are parameter-linear, these rules are Hebbian, only requiring the multiplication of pre and post-synaptic potentials. This means that the predictive coding updates proposed here are biologically plausible and could in theory be implemented in the brain. The only biological implausibility remains the BPTT learning scheme.

Our RNN was trained on a simple character-level name-origin dataset which can be found here: *https://download.pytorch.org/tutorial/data.zip*. The RNN was presented with sequences of characters representing names and had to predict the national origin of the name – French, Spanish, Russian, etc. The characters were presented to the network as one-hot-encoded vectors without any embedding. The output categories were also presented as a one-hot vector. The RNN has a hidden size of 256

units. A *tanh* nonlinearity was used between hidden states and the output layer was linear. The network was trained on randomly selected name-category pairs from the dataset. The training loss for the predictive coding and backprop RNNs, averaged over 5 seeds is presented below (Figure 7).

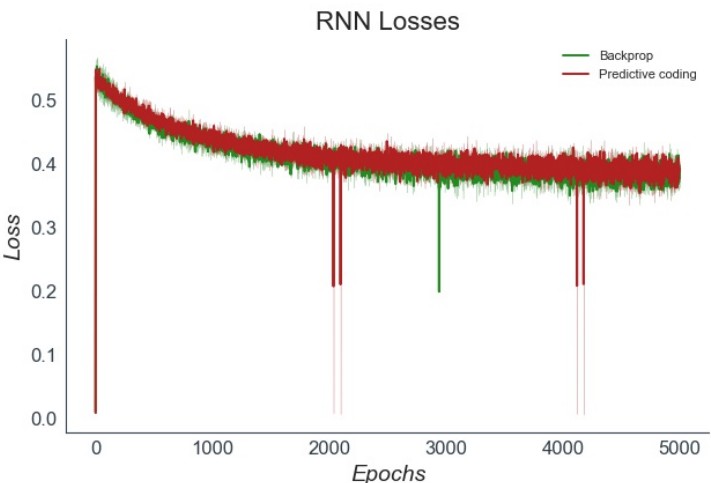

Figure 8: Training losses for the predictive coding and backprop RNN. As expected, they are effectively identical.

## APPENDIX C: PREDICTIVE CODING LSTM IMPLEMENTATION DETAILS

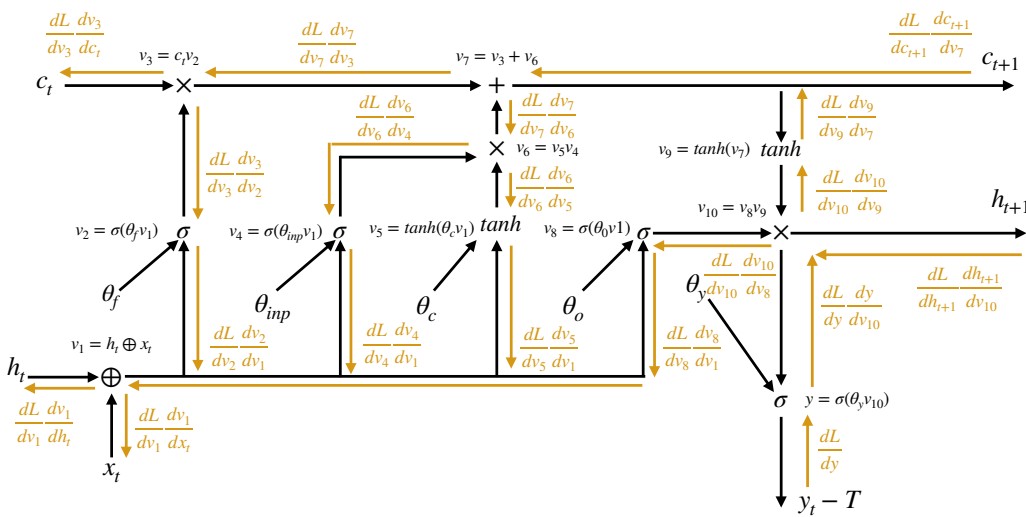

Figure 9: Computation graph and backprop learning rules for a single LSTM cell.

Unlike the other two models, the LSTM possesses a complex and branching internal computation graph, and is thus a good opportunity to make explicit the predictive coding 'recipe' for approximating backprop on arbitrary computation graphs. The computation graph for a single LSTM cell is shown (with backprop updates) in Figure 8. Prediction for the LSTM occurs by simply rolling forward a copy of the LSTM cell for each timestep. The LSTM cell receives its hidden state $h_t$ and cell state $c_t$ from the previous timestep. During training we compute derivatives on the unrolled computation graph and receive backwards derivatives (or prediction errors) from the LSTM cell at time $t+1$.

The equations that specify the computation graph of the LSTM cell are as follows.

$$v_1 = h_t \oplus x_t$$
$$v_2 = \sigma(\theta_i v_1)$$
$$v_3 = c_t v_2$$
$$v_4 = \sigma(\theta_{inp} v_1)$$
$$v_5 = \tanh(\theta_c v_1)$$
$$v_6 = v_4 v_5$$
$$v_7 = v_3 + v_6$$
$$v_8 = \sigma(\theta_o v_1)$$
$$v_9 = \tanh(v_7)$$
$$v_{10} = v_8 v_9$$
$$y = \sigma(\theta_y v_{10})$$

The recipe to convert this computation graph into a predictive coding algorithm is straightforward. We first rewire the connectivity so that the predictions are set to the forward functions of their parents. We then compute the errors between the vertices and the predictions.

$$\hat{v}_1 = h_t \oplus x_t$$
$$\hat{v}_2 = \sigma(\theta_i v_1)$$
$$\hat{v}_3 = c_t v_2$$
$$\hat{v}_4 = \sigma(\theta_{inp} v_1)$$
$$\hat{v}_5 = \tanh(\theta_c v_1)$$
$$\hat{v}_6 = v_4 v_5$$
$$\hat{v}_7 = v_3 + v_6$$
$$\hat{v} = \sigma(\theta_o v_1)$$
$$\hat{v}_9 = \tanh(v_7)$$
$$\hat{v}_{10} = v_8 v_9$$
$$\hat{v}_y = \sigma(\theta_y v_{10})$$
$$\epsilon_1 = v_1 - \hat{v}_1$$
$$\epsilon_2 = v_2 - \hat{v}_2$$
$$\epsilon_3 = v_3 - \hat{v}_3$$
$$\epsilon_4 = v_4 - \hat{v}_4$$
$$\epsilon_5 = v_5 - \hat{v}_5$$
$$\epsilon_6 = v_6 - \hat{v}_6$$
$$\epsilon_7 = v_7 - \hat{v}_7$$
$$\epsilon_8 = v_8 - \hat{v}_8$$
$$\epsilon_9 = v_9 - \hat{v}_9$$
$$\epsilon_{10} = v_{10} - \hat{v}_{10}$$

During inference, the inputs $h_t$,$x_t$ and the output $y_t$ are fixed. The vertices and then the prediction errors are updated according to Equation 2. This recipe is straightforward and can easily be extended to other more complex machine learning architectures. The full augmented computation graph, including the vertex update rules, is presented in Figure 9.

Empirically, we observed rapid convergence to the exact backprop gradients even in the case of very deep computation graphs (as is an unrolled LSTM with a sequence length of 100). Although convergence was slower than was the case for CNNs or lesser sequence lengths, it was still straightforward to achieve convergence to the exact numerical gradients with sufficient iterations.

Below we plot the mean divergence between the predictive coding and true numerical gradients as a function of sequence length (and hence depth of graph) for a fixed computational budget of 200 iterations with an inference learning rate of 0.05. As can be seen, the divergence increases roughly

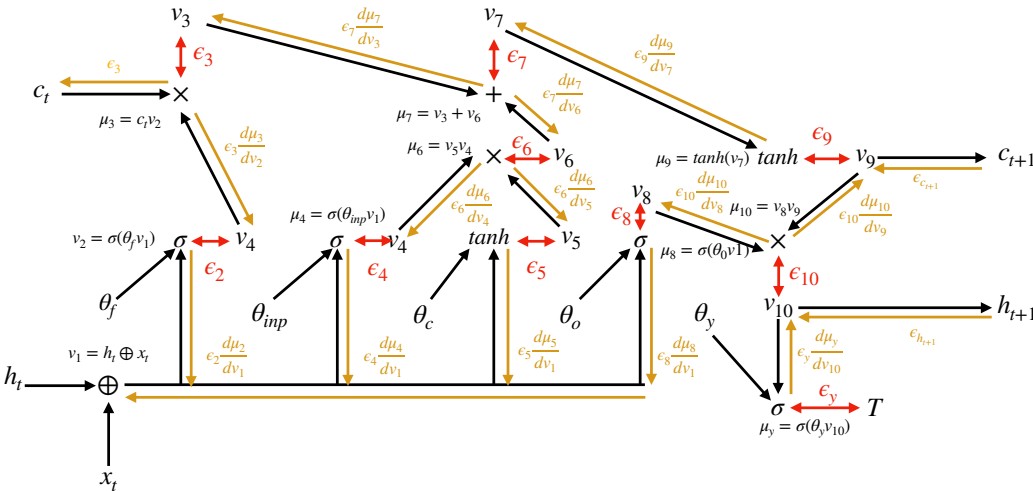

Figure 10: The LSTM cell computation graph augmented with error units, evincing the connectivity scheme of the predictive coding algorithm.

linearly with sequence length. Importantly, even with long sequences, the divergence is not especially large, and can be decreased further by increasing the computational budget. As the increase is linear, we believe that predictive coding approaches should be scalable even for backpropagating through very deep and complex graphs.

We also plot the number of iterations required to reach a given convergence threshold (here taken to be 0.005) as a function of sequence length (Figure 11). We see that the number of iterations required increases sublinearly with the sequence length, and likely asymptotes at about 300 iterations. Although this is a lot of iterations, the sublinear convergence nevertheless shows that the method can scale to even extremely deep graphs.

Our architecture consisted of a single LSTM layer (more complex architectures would consist of multiple stacked LSTM layers). The LSTM was trained on a next-character character-level prediction task. The dataset was the full works of Shakespeare, downloadable from Tensorflow. The text was shuffled and split into sequences of 50 characters, which were fed to the LSTM one character at a time. The LSTM was trained then to predict the next character, so as to ultimately be able to generate text. The characters were presented as one-hot-encoded vectors. The LSTM had a hidden size and a cell-size of 1056 units. A minibatch size of 64 was used and a weight learning rate of 0.0001 was used for both predictive coding and backprop networks. To achieve sufficient numerical convergence to the correct gradient, we used 200 variational iterations with an inference learning rate of 0.1. This rendered the predictive LSTM approximately 200x as costly as the backprop LSTM to run. A graph of the LSTM training loss for both predictive coding and backprop LSTMs, averaged over 5 random seeds, can be found below (Figure 12).

## APPENDIX D: DERIVATION OF THE FREE ENERGY FUNCTIONAL

Here we derive in detail the form of the free-energy functional used in sections 2 and 4. We also expand upon the assumptions required and the precise form of the generative model and variational density. Much of this material is presented with considerably more detail in Buckley et al. (2017), and more approachably in Bogacz (2017).

Given an arbitrary computation graph with vertices $\{y_i\}$, which we treat as random variables. Here we treat explicitly an important fact that we glossed over for notational convenience in the introduction. The $v_i$s which are optimized in the free-energy functional are technically the mean parameters of

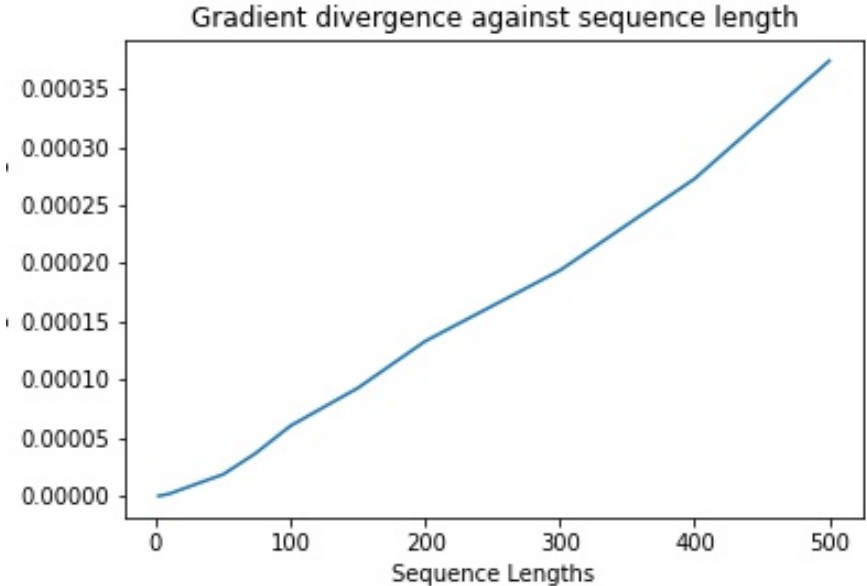

Figure 11: Divergence between predictive coding and numerical gradients as a function of sequence length.

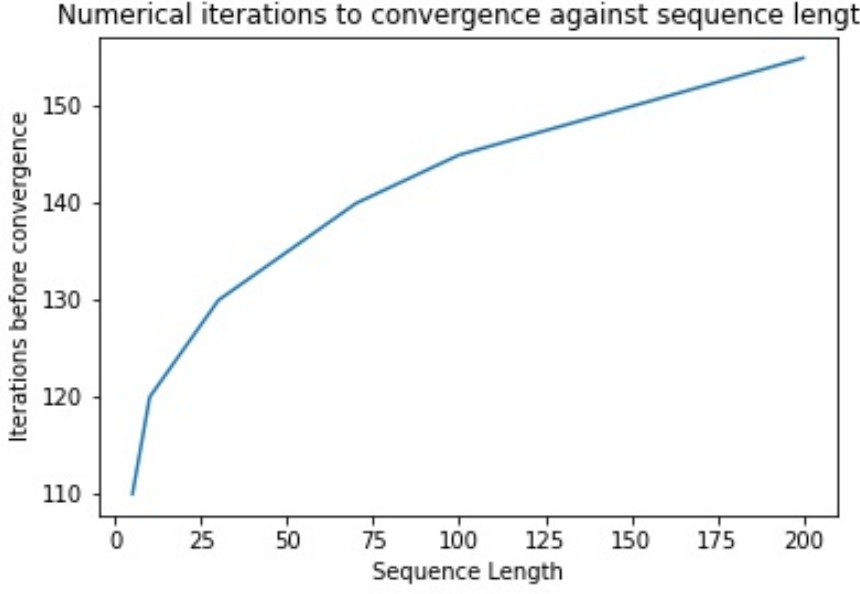

Figure 12: Number of iterations to reach convergence threshold as a function of sequence length.

the variational density $Q(y_i; v_i, \sigma_i)$ – i.e. they represent the mean (variational) belief of the value of the vertex. The vertex values in the model, which we here denote as $\{y_i\}$, are technically separate. However, due to our Gaussian assumptions, and the expectation under the variational density, in effect we end up replacing the $y_i$ with the $v_i$ and optimizing the $v_i$s, so in the interests of space and notational simplicity we began as if the $v_i$s were variables in the generative model, but they are not. They are parameters of the variational distribution.

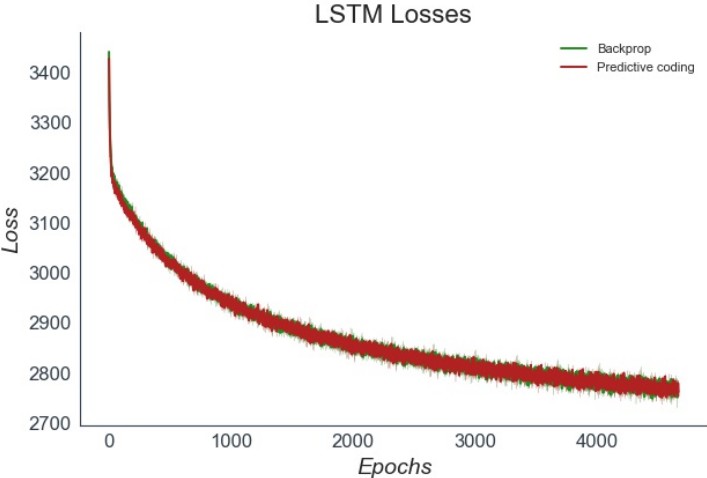

Figure 13: Training losses for the predictive coding and backprop LSTMs averaged over 5 seeds. The performance of the two training methods is effectively equivalent.

Given an input $y_0$ and a target $y_N$ (the multiple input and/or output case is a straightforward generalization). We wish to infer the posterior $p(y_{1:N-1}|y_0, y_N)$. We approximate this intractable posterior with variational inference. Variational inference proceeds by defining an approximate posterior $Q(y_{1:N-1}; \phi)$ with some arbitrary parameters $\phi$. We then wish to minimize the KL divergence between the true and approximate posterior.

$$\operatorname*{argmin}_{\phi} \mathbb{KL}[Q(y_{1:N-1}; \phi)\|p(y_{1:N-1}|y_0, y_N)]$$

Although this KL is itself intractable, since it includes the intractable posterior, we can derive a tractable bound on this KL called the variational free-energy.

$$
\begin{aligned}
\mathbb{KL}[Q(y_{1:N-1}; \phi)\|p(y_{1:N}|y_0, y_N)] &= \mathbb{KL}[Q(y_{1:N-1})\|\frac{p(y_{1:N}, y_0, y_N)}{p(y_0, y_N)}] \\
&= \mathbb{KL}[Q(y_{1:N}; \phi)\|p(y_{1:N}, y_0)] + \ln p(y_0, y_N) \\
&\Rightarrow \underbrace{\mathbb{KL}[Q(y_{1:N}; \phi)\|p(y_{1:N-1}, y_0, y_N)]}_{-\mathcal{F}} \leq \mathbb{KL}[Q(y_{1:N-1}; \phi)\|p(y_{1:N-1}|y_0, y_N)]
\end{aligned}
$$

(9)

We define the negative free-energy $-\mathcal{F} = \mathbb{KL}[Q(y_{1:N-1})\|p(y_{1:N-1}, y_0, y_N)]$ which is a lower bound on the divergence between the true and approximate posteriors. By thus maximizing the negative free-energy (which is identical to the ELBO (Beal et al., 2003; Blei et al., 2017)), or equivalently minimizing the free-energy, we decrease this divergence and make the variational distribution a better approximation to the true posterior.

To proceed further, it is necessary to define an explicit form of the generative model $p(y_0, y_{1:N-1}, y_N)$ and the approximate posterior $Q(y_{1:N-1}; \phi)$. In predictive coding, we define a hierarchical Gaussian generative model which mirrors the exact structure of the computation graph

$$p(y_{0:N}) = \mathcal{N}(y_0; \bar{y}_0, \Sigma_0) \prod_{i=1}^{N} \mathcal{N}(y_i; f(\mathcal{P}(y_i); \theta_{y_j \in \mathcal{P}(y_i)}), \Sigma_i);$$

Where essentially each vertex $y_i$ is a Gaussian with a mean which is a function of the prediction of all the parents of the vertex, and the parameters of their edge-functions. $\bar{y}_0$ is effectively an "input-prior" which is set to 0 throughout and ignored. The output vertices $y_N = T$ are set to the target $T$.

We also define the variational density to be Gaussian with mean $v_{1:N-1}$ and variance $\sigma_{1:N-1}$, but under a mean field approximation, so that the approximation at each node is independent of all others

(note the variational variance is denoted $\sigma$ while the variance of the generative model is denoted $\Sigma$. The lower-case $\sigma$ is not used to denote a scalar variable – both variances can be multivariate – but to distinguish between variational and generative variances)

$$Q(y_{1:N-1}; v_{1:N-1}, \sigma_{1:N-1}) = \prod_{i=1}^{N-1} \mathcal{N}(y_i; v_i, \sigma_i)$$

We now can express the free-energy functional concretely. First we decompose it as the sum of an energy and an entropy

$$-\mathcal{F} = \mathbb{KL}[Q(y_{1:N-1}; v_{1:N-1}, \sigma_{1:N-1}) \| p(y_0, y_{1:N-1}, y_N)]$$
$$= \underbrace{-\mathbb{E}_{Q(y_{1:N-1}; v_{1:N-1}, \sigma_{1:N-1})}[\ln p(y_0, y_{1:N-1}, y_N)]}_{Energy} + \underbrace{\mathbb{E}_{Q(y_{1:N-1}; v_{1:N-1}, \sigma_{1:N-1})}[\ln Q(y_{1:N-1}; v_{1:N-1}, \sigma_{1:N-1})]}_{Entropy}$$

Then, taking the entropy term first, we can express it concretely in terms of normal distributions.

$$\mathbb{E}_{Q(y_{1:N-1}; v_{1:N-1}, \sigma_{1:N-1})}[\ln Q(y_{1:N-1}; v_{1:N-1}, \sigma_{1:N-1})] = \mathbb{E}_{Q(y_{1:N-1}; v_{1:N-1}, \sigma_{1:N-1})}[\sum_{i=1}^{N-1} \ln \mathcal{N}(y_i; v_i, \sigma_i)]$$

$$= \sum_{i=1}^{N-1} \mathbb{E}_{Q(y_i; v_i, \sigma_i)}[\ln \mathcal{N}(y_i; v_i, \sigma_i)]$$

$$= \sum_{i=1}^{N-1} \mathbb{E}_{Q(y_i; v_i, \sigma_i)}[-\frac{1}{2}\ln\det(2\pi\sigma_i)] + \mathbb{E}_{Q(y_i; v_i, \sigma_i)}[\frac{(y_i - v_i)^2}{2\sigma_i}]$$

$$= \sum_{i=1}^{N-1} -\frac{1}{2}\ln\det(2\pi\sigma_i)] + \frac{\sigma_i}{2\sigma_i}$$

$$= \frac{N}{2} + \sum_{i=1}^{N-1} -\frac{1}{2}\ln\det(2\pi\sigma_i)$$

The entropy of a multivariate gaussian has a simple analytical form depending only on the variance. Next we turn to the energy term, which is more complex. To derive a clean analytical result, we must make a further assumption, the Laplace approximation, which requires the variational density to be tightly peaked around the mean so the only non-negligible contribution to the expectation is from regions around the mean. This means that we can successfully approximate the approximate posterior with a second-order Taylor expansion around the mean. From the first line onwards we ignore the $\ln p(y_0)$ and $\ln p(y_N|\mathcal{P}(y_N))$ which lie outside the expectation.

$$\mathbb{E}_{Q(y_{1:N-1}; v_{1:N-1}, \sigma_{1:N-1})}[\ln p(y_{0:N})] = \ln p(y_0) + \ln p(y_N|\mathcal{P}(y_N)) + \sum_{i=1}^{N-1} \mathbb{E}_{Q(y_i; v_i, \sigma_i)}[\ln p(y_i|\mathcal{P}(y_i))]$$

$$= \sum_{i=1}^{N} E_Q[\ln p(v_i|\mathcal{P}(y_i))] + \mathbb{E}_Q[\frac{\partial \ln p(y_i|\mathcal{P}(y_i))}{\partial y_i}(v_i - y_i)]$$

$$+ \mathbb{E}_Q[\frac{d^2 \ln p(v_i|\mathcal{P}(y_i))}{dy_i^2}(v_i - y_i)^2]$$

$$= \sum_{i=1}^{N} \ln p(v_i|\mathcal{P}(y_i)) + \frac{\partial^2 \ln p(v_i|\mathcal{P}(y_i))}{\partial y_i^2}\sigma_i$$

Where the second term in the Taylor expansion evaluates to 0 since $\mathbb{E}_Q[y_i - v_i] = (v_i - v_i) = 0$ and the third term contains the expression for the variance $\mathbb{E}_Q[(y_i - v_i)^2] = \sigma_i$.

We can then write out the full Laplace-encoded free-energy as:

$$-\mathcal{F} = \sum_{i=1}^{N} \ln p(v_i|\mathcal{P}(y_i)) + \frac{\partial^2 \ln p(v_i|\mathcal{P}(y_i))}{\partial y_i^2}\sigma_i - -\frac{1}{2}\ln\det(2\pi\sigma_i)$$

We wish to minimize $\mathcal{F}$ with respect to the variational parameters $v_i$ and $\sigma_i$. There is in fact a closed-form expression for the optimal variational variance which can be obtained simply by differentiating and setting the derivative to 0.

$$\frac{\partial \mathcal{F}}{\partial \sigma_i} = \frac{\partial^2 \ln p(v_i | \mathcal{P}(y_i))}{\partial y_i^2} - \sigma_i^{-1}$$

$$\frac{\partial \mathcal{F}}{\partial \sigma_i} = 0 \Rightarrow \sigma_i^* = \frac{\partial^2 \ln p(v_i | \mathcal{P}(y_i))}{\partial y_i^2}^{-1}$$

Because of this analytical result for the variational variance, we do not need to consider it further in the optimisation problem, and only consider minimizing the variational means $v_i$. This renders all the terms in the free-energy except the $\ln p(v_i | \mathcal{P}(y_i))$ terms constant with respect to the variational parameters. This allows us to write:

$$-\mathcal{F} \approx \ln p(y_N | \mathcal{P}(y_N)) + \sum_{i=1}^{N} \ln p(v_i | \mathcal{P}(y_i)) \tag{10}$$

as presented in section 2. The first term $\ln p(y_N | \mathcal{P}(y_N))$ is effectively the loss at the output ($y_N = T$) so becomes an additional prediction error $\ln p(y_N | \mathcal{P}(y_N)) \propto (T - \hat{v}_N)^T \Sigma_N^{-1} (T - \hat{v}_N)$ which can be absorbed into the sum over other prediction errors. Crucially, although the variational variances have an analytical form, the variances of the generative model (the precisions $\Sigma_i$) do not and can be optimised directly to improve the log model-evidence. These precisions allow for a kind of 'uncertainty-aware' backprop.

DERIVATION OF VARIATIONAL UPDATE RULES AND FIXED POINTS

Here, starting from Equation 10, we show how to obtain the variational update rule for the $v_i$'s (Equation 2), and the fixed point equations (Equation 5) (Friston, 2008; 2005; Bogacz, 2017). We first reduce the free-energy to a sum of prediction errors.

$$-\mathcal{F} \approx \sum_{i=1}^{N} \ln p(v_i | \mathcal{P}(v_i))$$

$$\approx \sum_{i=1}^{N} (v_i - f(\mathcal{P}(v_1))^T \Sigma_i^{-1} (v_i - f(\mathcal{P}(v_1))^T + \ln 2\pi \Sigma_i^{-1}$$

$$= \sum_{i=1}^{N} \epsilon_i^T \epsilon_i + \ln 2\pi \Sigma_i^{-1}$$

Where $\epsilon_i = v_i - f(\mathcal{P}(v_1))$, and we have utilized the assumption made in section 2 that $\Sigma^{-1} = \mathbf{I}$. By setting all precisions to the identity, we are implicitly assuming that all datapoints and vertices of the computational graph have equal variance. Next we assume that the dynamics of each vertex $v_i$ follow a gradient descent on the free-energy.

$$-\frac{dv_i}{dt} = \frac{\partial \mathcal{F}}{\partial v_i} = \frac{\partial}{\partial v_i} [\sum_{j=1}^{N} \epsilon_j^T \epsilon_j]$$

$$= \epsilon_i \frac{\partial \epsilon_i}{\partial v_i} + \sum_{j \in \mathcal{C}(v_i)} \epsilon_j \frac{\partial \epsilon_j}{\partial v_i}$$

$$= \epsilon_i - \sum_{j \in \mathcal{C}(v_i)} \epsilon_j \frac{\partial \hat{v}_j}{\partial v_i}$$

Where we have used the fact that $\frac{\partial \epsilon_i}{\partial v_i} = 1$ and $\frac{\partial \epsilon_j}{\partial v_j} = -\frac{\partial \hat{v}_j}{\partial v_i}$. To obtain the fixed point of the dynamics, we simply solve for $\frac{dv_i}{dt} = 0$.

$$\frac{dv_i}{dt} = \frac{\partial \mathcal{F}}{\partial v_i} = 0$$

$$\Rightarrow 0 = \epsilon_i - \sum_{j \in \mathcal{C}(v_i)} \epsilon_j \frac{\partial \hat{v}_j}{\partial v_i}$$

$$\Rightarrow \epsilon_i^* = \sum_{j \in \mathcal{C}(v_i)} \epsilon_j \frac{\partial \hat{v}_j^*}{\partial v_i^*}$$

Similarly, since $\epsilon_i^* = v_i^* - \hat{v}_i^*$ then $v_i^* = \epsilon_i^* + \hat{v}_i^*$. So:

$$v_i^* = \epsilon_i^* + \hat{v}_i^*$$

$$= \hat{v}_i^* - \sum_{j \in \mathcal{C}(v_i)} \epsilon_j \frac{\partial \hat{v}_j^*}{\partial v_i^*}$$

