# OpenReview forum: "Predictive Coding Approximates Backprop along Arbitrary Computation Graphs"
_ICLR.cc/2021/Conference — Reject_

### Official Review · AnonReviewer4 · 2020-10-24
**The paper is not clear enough**

**Rating:** 4
**Confidence:** 4

**Review:**

#### Summary of the paper
In their paper, the authors demonstrate that Predictive Coding (PC) is a local approximation of back-propagation and could then be interpreted with Hebbian learning rule (a neuro-plausible learning rule). This result has been first demonstrated by [1] with MLP network (on the MNIST dataset) and the presented paper extend this finding to CNNs (on CIFAR10, CIFAR100 and SVHN), RNN and LSTM.

#### Pros
* The authors provided experimental evidences on a wide variety of networks' type and databases.
* The link between neuro-plausible learning rule and back propagation is interesting.
* The paper is well situated in the literature.
* The authors are providing the code for clean reproducibility

#### Cons
* The mathematical definition and notation of the paper are not rigorous enough. It makes the paper unclear and hard to follow.
* Some crucial points would have deserved in-depth discussion and are just ignored (see below)
* The paper is not well enough motivated: what’s the point of such a local approximation beside the neuro-plausibility (faster ? Consume less resources ? …)
* The demonstration seems to include only one kind of loss function, which does not match the claim of the paper

#### Recommendation
Given the limited impact and the lack of clarity of the paper, I would tend to reject the article.

#### Detailed comments:
* The gaussian parametrization used by the authors constrains the comparison between PC and backprop to networks with L2 loss function. One cannot claim to approximate arbitrary computational graph if one demonstrates the approximation on a specific loss function (which is known to poorly perform on classification problem). So could your framework be generalized to more effective loss function like cross entropy ? If yes, what would be the underlying probabilistic hypothesis ? This should be included in the paper, as it will strongly strengthen your claim.

* The PC framework proposed by the authors propose a solution to the ’non locality’ of the back propagation (to be a bio-plausible mechanism). However the authors also raised the weight transport problem. On my understanding the proposed framework is still suffering from weight transport as the backward connection weights are the transpose of the feedforward one (due to the derivation of the forward operator). The paper would deserve an in-depth discussion concerning this point.

* What is the computational advantage of local approximations ? Is it saving computational resources (computational time, memory…) ? A comparison of the algorithmic complexity between PC and back-propagation would be valuable to support your claim. In the discussion, the authors mention that their framework, being substantially more expensive than back-prop network, could be deeply parallelized across layer. The authors should provide experimental or theoretical evidences that such parallelization is enough to mitigate the higher number of inference steps (i.e. 100-200) needed by their PC framework.

* The concept of ‘fixed-prediction assumption’ introduced by authors in the paper considers that each (v_i) are fixed to their feedforward value. Then what is the point of the Eq. 2, as you already know the value of the activity vector? On my mind this is here a crucial point, as this is dealing with the core principle of PC : an inner-loop (i.e. the expectation step) that find the most likely v_i, and an outer loop (i.e. the maximization step) that update the parameters. I have the intuition that this problem arises because the authors are tackling a discriminative problem (i.e. finding a mapping between inputs and labels) using a generative model (PC is a generative generative model as described by [2, 3]). Can you please clarify this point ?

* What is the testing procedure of your network ? Is it a simple feedforward pass (which I suspect) or is it an Expectation-Maximization scheme. Your algorithm 1 shows only the training procedure (as you need the label information to perform the computation). If this is a simple feedforward pass, what would be the advantages of the inference process (more robustness ? Better prediction ?)

## Typos and suggestions to improve the paper :
* The authors state (3rd paragraph page 3) that they are considering a generative model. If it is the case, the formula p(v_i) = product(p(v_i | parent(v_i)) is inaccurate as the authors forgot the prior p(v_N).
* Eq 1 : The derivation between the first and the second line of Eq.1 has to be demonstrated or referenced (at least in annex)…
* The authors consider the posterior is a marginal probability (see eq1, and subsequent paragraph). In general the posterior is a conditional probability (this specific point makes your equation 1 hard to grasp because readers are not making the link with the classical ELBO, i.e. the negative free energy). In general, the probabilistic notations are not rigorous enough, and it makes the rest of the mathematical derivation complicated to follow.
* The authors should reorganize the Annex to make sure it follows the reading order (Appendix D is cited first)
* Caption of Figure 1 : backwawrds —> backwards
* Page 3, 2 lines below eq 1 : as as —> as
* Page 4, 4 lines below eq 3 : forwards —> forward
* Figure 3, which is on my mind the most import one, is shown but not cited in the core text.

[1] Whittington, J. C., & Bogacz, R. (2017). An approximation of the error backpropagation algorithm in a predictive coding network with local hebbian synaptic plasticity. Neural computation, 29(5), 1229-1262.

[2] Rao, Rajesh PN, and Dana H. Ballard. "Predictive coding in the visual cortex: a functional interpretation of some extra-classical receptive-field effects." Nature neuroscience 2.1 (1999): 79-87.

[3]Friston, Karl. "A theory of cortical responses." Philosophical transactions of the Royal Society B: Biological sciences360.1456 (2005): 815-836.

---

> ### Author Response · Authors · 2020-11-20
> **Response to reviewer 4**
>
> We thank the reviewer for their extremely detailed and thorough review, which will undoubtedly help us improve the manuscript, and also for the many typos and minor mistakes they have highlighted which we are happy to fix.
>
> Below are responses to the detailed comments:
>
> 1.) The gaussian generative model does assume a mean-square-error loss function. This does not mean that other loss functions cannot be used. If you want to use another loss function that can be represented as the log of a distribution, then the final output distribution p(v_N | v_N-1) can be set to the distribution necessary to represent the other loss function. If the loss function cannot be represented as the log of a conditional distribution, then the predictive coding algorithm (i.e. equations 2 and 3) will still work, except with the algorithm as a whole will no longer have the elegant interpretation as variational inference on a specific generative model. It is important to note that all the other nodes except the output node can still be gaussian and optimize a mean-square-error regardless of the choice of loss function at the output layer. Nevertheless, this is a good point and we have added a footnote to this effect in the revised version of the paper.
>
> 2.) The reviewer is correct that this current proposal of using predictive coding does not address the weight transport problem -- only the issues of nonlocality and sequential computation inherent in backprop. In the literature there are already several proposed remedies for the weight transport problem such as Feedback Alignment (Lillicrap 2016) and learning the backwards weights (Amit 2019) which could equally well be used in the predictive coding algorithm as compared to backprop networks where they have currently been tested. Indeed, more recent work (https://arxiv.org/pdf/2010.01047.pdf), has shown that the weight transport problem can be addressed in the predictive coding framework through a set of independent learnable backwards weights which can be trained with a Hebbian learning rule. In a revised draft (to follow soon), we shall include a section discussing the overall biological plausibility of the algorithm and linking to these papers.
>
> We have added the following paragraph discussing this into the related work section of the revised version of the paper (to follow shortly):
>
> """"""
> It is important to note that predictive coding, as advanced here, still retains some biologically implausible features. Although using only local and Hebbian updates, the predictive coding algorithm still requires identical forward and backwards weights, as well as mandating a very precise one-to-one connectivity structure between value neurons $v_i$ and error neurons $\epsilon_i$. However, recent work () has begun to show that these implausibilities can be relaxed using learnable backwards weights instead of requiring weight symmetry, allowing for learnable dense connectivity between value and error neurons, without harm to performance in simple MLP settings.
> """"""
>
> 3.) The main advantage of the locality of the algorithm is theoretical -- that it allows for a parallel implementation and, crucially, is a step closer to the kind of biologically plausible credit assignment and learning that could take place in the brain. We believe that this is of intrinsic intellectual interest, although it may also prove useful computationally to aid in the construction and design of neuromorphic hardware, or in training deep neural networks in highly parallel settings. In theory the parallel properties of this algorithm could allow for it to be faster than backprop in a sufficiently parallel setup, potentially on neuromorphic hardware. However, this is an avenue for future work and out of scope for this paper.

---

> > ### Author Response · Authors · 2020-11-20
> > **Response to reviewer 4 -- continued**
> >
> > Continued from before...
> >
> > 4.) The reason that the inner-loop optimisation problem is not trivial given the fixed prediction assumption, is that the update rule for the $v_i$ depends not only on the current layer (which would be trivial) but also the layer above. This means that information and dependencies flows between layers, so that the total (across all layers) sum of prediction errors is minimized rather than just each layer’s prediction error  independently. Crucially, at the output layer the real labels are fed into the network and generates a prediction error which cannot be trivially minimized. Over the course of the optimization, this output prediction error is slowly spread across all nodes in the network due to the operation of the inner-loop optimisation rule until it becomes equivalent to the backpropagation gradients at convergence.
> >
> > As far as the authors are aware, this does not relate to the generative/discriminative model distinction. Importantly, the predictive coding network, although tested in a discriminative setting, is actually still a generative model. Simply one where the "discriminative" direction -- generating labels from data is straightforward. It is also possible to run the network “backwards” so that the labels are fixed and the inner-loop will optimize a prediction of the data (i.e. reconstruct the image).
> >
> > 5.) The inference process in our network is just a simple feedforward pass. In the inference phase, the predictive coding network is equivalent to a standard ANN. The key difference is that it can be trained with a local learning algorithm instead of backprop. If the network is run “backwards” to generate data from the labels, then the inference would be done through the E-step of an EM scheme.
> >
> > Specific comments on typos and suggestions:
> > 1.) Regarding the point that the generative model does not include the prior p(v_N) -- this is technically not true since the generative model description is defined as p(v_i | parents(v_i)) for every node, including the prior. It is just that the parents of the prior is the empty set, giving us just p(v_N). Nevertheless, we shall definitely clarify this in the revised version of the paper. Moreover, in derivations in Appendix D, we do explicitly include the prior.
> >
> > 2.) The derivation of line 2 of Equation 1 from line 1 is provided in Appendix D, as well as the derivation of the update rules (Equations 2 and 3)
> >
> > 3.) We do consider the posterior to be a conditional distribution. We denote the variational distributions Q(x) as not an explicit conditional distributions, since they only depend on the data through an optimization process, not directly (as in an amortised encoder). If this causes confusion, we are happy to change the notation. We always denote the true posterior ( p(v_{1...N | v_0, v_N) ) as a conditional distribution. Perhaps the confusion has arisen from the fact that the first equality in Equation 1 is the ELBO and we represent the joint as p({v_i}) which looks like a marginal distribution?
> > 4.) You make a really good point about the organization of the appendix. We shall reorder the appendix in the revised version to reflect the order in which the sections are cited in the main text.
> >
> > We hope that this response addresses some of your concerns -- thanks again for the really detailed response. It will definitely help us improve the manuscript going forwards.

---

> > ### Comment · AnonReviewer4 · 2020-11-21
> > **Response to authors**
> >
> > 1) I think this point is actually very important as this would support the authors claim that their algorithm could approximate any computational graph (and not only graph based on L2 loss). However, an experimental verification of the authors suggestion (i.e. replacing the last layer probability density by a categorical distribution, in the case of the categorical cross entropy) is necessary to illustrate and confirm this point.

---

> > > ### Author Response · Authors · 2020-11-23
> > > **response to reviewer 4 continued**
> > >
> > > We agree with the reviewer that demonstrating that the algorithm works with alternative loss functions (such as crossentrpy) is important to show the generality of the method. In the revised version (now up), we have added experiments comparing predictive coding and backprop with the crossentropy with the CNN architecture (Figure 7) and shown that performance closely matches that of the backprop network.

---

### Official Review · AnonReviewer3 · 2020-10-27
**Clear paper - marginal originality**

**Rating:** 6
**Confidence:** 4

**Review:**

##########################################################################

Summary:

Authors propose that predictive coding gives similar convergence as backprop algorithms by extending the work of (Whittington &Bogacz 2017,  https://www.mrcbndu.ox.ac.uk/sites/default/files/pdf_files/Whittington%20Bogacz%202017_Neural%20Comput.pdf) to arbitrary graphs. This is an important topic both for the application of classical deep nets to neuromorphic hardware, but also for our understanding of computations in biological tissues.

1. The work presented in this paper follows similar results by Amit (2019) or Lilicrap, and provides with numerical simulations comforting the theoretical predictions.
2. Authors present an extension of the previous work to arbitrary graphs, and they apply their claims to LSTM and RNN models.
3. It provides comprehensive experiments, including both qualitative analysis and quantitative results, to show the effectiveness of the proposed framework. This makes the point of the paper more convincing and complementary to similar works.

##########################################################################

Concern:

A major issue of this paper is to not identify its originality. The extension to « arbitrary graphs » or to CNNs is straightforward in theory, While it is not explicitly stated in  (Whittington & Bogacz, 2017), an unwrapped RNN is by definition a feed-forward graph and is therefore a direct application of their work.

Concerning novelty, the actual derivations (eg for the LSTM) are original and the simulations clearly support that original contributions. However this material is at the end of the paper or in appendices. Recentered on this original contributions and how this makes a suitable contribution to the community would make the paper acceptable to be accepted at ICLR.

##########################################################################
minor:
p.2 « backwawrds »
p7 « dataest » & p 14

---

> ### Author Response · Authors · 2020-11-20
> **Response to reviewer 3**
>
> We thank the reviewer for their insightful comments. Regarding the originality, we still believe that the extension to arbitrary graphs (including CNNs and RNNs) instead of just a MLP network is not completely trivial and has not been shown before in the literature. We agree that it is unfortunate that the derivations for the CNN, LSTM etc are in the appendix, which is done primarily for space constraints. If space permits, we will be happy to move some of this material into the main text.

---

### Official Review · AnonReviewer2 · 2020-10-28
**Generalisation of predictive coding approach to arbitrary computational graphs could be quite powerful**

**Rating:** 6
**Confidence:** 4

**Review:**

### Update after author responses:
While the author address some of my comments, I would have still liked to see a more detailed discussion of how the algorithm compares in terms of algorithmic scaling, which I think is relevant because it is a fundamental property of the algorithm, even if it is targeted towards understanding biology. So my score remains the same.

Summary:

The authors extend recent work on MLPs to show that predictive coding converges asymptotically to exact backprop gradients on arbitrary computation graphs. They construct predictive coding networks for common architectures and show that it works well.

Overall, I vote for an accept because I think the generalisation is quite useful and interesting for training deep networks with local learning rules. The authors demonstrate that this method works, but haven't demonstrated its computational advantages clearly enough. There are some issues of clarity that I have also outlined below.

Strengths:
+ The generalisation of the earlier MLP results to arbitrary computational graphs is quite powerful esp. since it can be applied to most deep learning architectures.
+ The experimental evaluation includes all popular deep learning architecture, and it's impressive that this works on all of them. The experimental evaluation is also extensive.

Weaknesses:
- The increase in computational cost (of 100x) is mentioned quite late and seems to be glossed over a bit.
- Due to the potential for parallelisation in the predictive coding network, a comparison of wall-clock time for training on highly parallel setups might have been very interesting.
- For RNNs and LSTMs, the equivalent predictive coding network is generated after unrolling the network, which means that the predictive coding network has no memory advantage over BPTT and a huge performance penalty. This is related to the previous point, where the utility of the predictive coding network is not demonstrated sufficiently.

Clarity:
- Many of the figures are almost unreadable on paper. E.g. Fig. 3.
- Algorithm 1 does't seem to be referenced anywhere.
- In fig. 1 bottom, $\delta$ missing in the denominators
- Eqn. 3 is a bit sloppy, where derivative w.r.t $\theta$ is suddenly equated to a derivative w.r.t $\theta_i$.
- If $\epsilon_i = v_i - \hat{v}_i$ then eqn. 7 is inconsistent with this, since $\frac{d\epsilon^*_i}{d\theta}$ would be $-\frac{d\hat{v}_i}{d\theta_i}$ (missing -ve sign). Unless I misunderstood something.

---

> ### Author Response · Authors · 2020-11-20
> **First response to reviewer 2**
>
> We thank the reviewer for their helpful and insightful review, and also thank them for their comments on clarity and alerting us to the various typos. Regarding Eqn 3 and 7, this was indeed an oversight on our part -- all the $\theta$s should be $\theta_i$s and the -ve sign is missing. We will fix this in the revised version (to follow soon), and also increase the size of the figures, given the extra page, especially Figure 3.
>
> Regarding the additional computational cost, 100x provides an upper bound. This figure is for (almost) complete convergence with a relatively low learning rate. Preliminary experiments indicate (as can also be seen by eyeballing figure 2 (left) that convergence can require fewer iterations with higher learning rates -- often about 10-20 iterations in practice. Nevertheless, we do not envisage the contribution of this paper to be a practical algorithm directly competitive with backprop in terms of computational cost. Instead, we have presented a fully parallelizable and local approximation to backprop that has a number of interesting and biologically plausible properties. We hope that demonstrating these properties inspires the development of more efficient algorithms in the future, and helps contextualise research into backpropagation and the brain. However, we do agree that testing the inherent parallelism of the algorithm both in a cluster environment as well as on neuromorphic hardware would be a very interesting project, although out of scope for the current paper.
>
> We also agree about predictive coding on recurrent networks like RNNs or LSTMs not being directly useful in terms of memory or particularly in terms of biological plausibility either. The aim in showcasing these networks was primarily a proof-of-concept to show that predictive coding could be applied on very complex graphs like that of an unrolled LSTM rather than as a competitive algorithm.

---

### Official Review · AnonReviewer1 · 2020-10-28
**Prective coding shown to converge to backprop gradients for abritrary computational graphs**

**Rating:** 7
**Confidence:** 3

**Review:**

The paper extends prior work on equivalence between predictive coding and backprop in layered neural networks to arbitrary computation graphs. This is empirically tested first on a simple nonlinear scalar function, and then on a few commonly used architectures (CNNs, RNNs, LSTMs), confirming the theoretical results. The importance of this advance is highlighted by noting that the demonstrated equivalence shows how in principle modern architectures could be implemented in biological neural systems, and that the highly parallel nature of predictive coding could lead to efficient implementations in neuromorphic hardware.

The paper is very well written, easy to read, and includes a nice introduction section with a fairly comprehensive overview of backprop, and the problems related to its potential implementations in biological systems. I also appreciated the "tension in a chain" metaphor illustrating the dynamics of backprop and predictive coding. That the exact backprop gradients are computable in a fully local system with Hebbian plasticity for an arbitrary graph is an interesting and promising result.

Throughout the text, predictive coding is quoted as biologically plausible. This isn't strictly true as noted already in (Whittington & Bogacz, 2017), as e.g. dedicated error nodes are not known to exist for every cell in the brain. I'd suggest calling this "potentially biologically plausible", and including a short discussion on how these plausibility concerns could be addressed.

All in all, the results are interesting, open up interesting directions for future work, and I recommend the acceptance of the paper.

Additional questions/suggestions:
- Is the fixed-prediction assumption a limitation to biological plausibility?
- Fig. 2 shows the model converging at high inference learning rates for the case of the scalar function. Is the 0.1 rate used for CNNs the max that was stable, or could higher values be used to reduce the computational overhead?
- What convergence condition was used?
- How does convergence speed depend on the size/diameter of the computational graph? This is similar to Fig. 9, but asking a slightly different question -- i.e. how many iterations are needed to reach convergence as a function of graph size.

Typos:
Fig. 1 caption: "backwawrds"

---

> ### Author Response · Authors · 2020-11-20
> **First response to reviewer 1**
>
> We thank the reviewer for their detailed review and positive appraisal. With regards to softening  the claim of biological plausibility to “potentially biologically plausible”, we agree with this point and intend to include a more detailed discussion of this in the revised paper (to follow shortly). Specifically, we will discuss the status of the dedicated error nodes, the symmetric backwards and forward weights, and the fixed-prediction assumption (which is likely a limitation of biological plausibility). Also note that there has been recent work showing that some of the less biologically plausible aspects of predictive coding -- such as symmetric weights and  one-to-one error neuron to value neuron connectivity can be removed without significantly  harming the performance of the algorithm (https://arxiv.org/pdf/2010.01047.pdf) in simple MLP architectures. We have added a short discussion of this in the related work section of the revised version of the paper.
>
> Specific responses:
> 1.)  The fixed-prediction assumption as stated is a limitation of biological plausibility. However, it is also important to note that the fixed-prediction assumption is only required for exact convergence to the gradients computed by backprop. Predictive coding without this assumption has been shown to attain very good classification performance (Whittington and Bogacz 2017) without this assumption, but no longer converges exactly to backprop. We anticipate that a good avenue for future work would lie in figuring out whether this assumption can be relaxed, or the design of predictive coding inspired algorithms which do not require it. The fixed-prediction assumption is required for exact convergence to the backpropagated gradients because for the gradients to match the output of the network must match the forward pass of the corresponding ANN upon which backprop is performed. Without the fixed-prediction assumption, the outputs of the network are allowed to evolve during convergence, thus leading to different gradients being computed.
>
> 2.)  Above 0.1 the update rules can become unstable (0.1 is already very high as a euler-integration time-step for an ODE). This happens especially in the long computation graphs of the LSTM. CNNs can typically tolerate learning rates up to about 0.2-0.3 without serious danger of instability.
>
> 3.) For the experiments in this paper we did not use a convergence condition but ran the predictive coding update for 100 iterations,which we found empirically to yield very good convergence in almost all cases. This was further verified by comparing the estimated predictive coding gradients against the true backprop gradients (as in Figure 5).  In practice, a convergence condition can be constructed in a principled manner by using the variational free energy (i.e. the sum of prediction errors). However, the aim of the paper was to establish and verify the connection between predictive coding and backprop, rather than optimize for computational overhead.
>
> 4.) We investigated the number of iterations required to reach a certain level of convergence and found a roughly sublinear relationship between graph size and iterations, similar to Figure 9. We have included this graph in the revised version of the manuscript. We found the relationship to be somewhat sublinear, so that the number of iterations required to reach a specific convergence threshold does not scale linearly with computation graph size but asymptotes at about 200-300 iterations.
>
> We hope these points help address and clarify the reviewer's comments and concerns

---

### Decision · Program_Chairs · 2021-01-07
**Final Decision**

**Decision:**

Reject

**Comment:**

This paper extends recent work (Whittington & Bogacz, 2017, Neural computation, 29(5), 1229-1262) by showing that predictive coding (Rao & Ballard, 1999, Nature neuroscience 2(1), 79-87) as an implementation of backpropagation can be extended to arbitrary network structures. Specifically, the original paper by Whittington & Bogacz (2017) demonstrated that for MLPs, predictive coding converges to backpropagation using local learning rules. These results were important/interesting as predictive coding has been shown to match a number of experimental results in neuroscience and locality is an important feature of biologically plausible learning algorithms.

The reviews were mixed. Three out of four reviews were above threshold for acceptance, but two of those were just above. Meanwhile, the fourth review gave a score of clear reject. There was general agreement that the paper was interesting and technically valid. But, the central criticisms of the paper were:

1) Lack of biological plausibility
The reviewers pointed to a few biologically implausible components to this work. For example, the algorithm uses local learning rules in the same sense that backpropagation does, i.e., if we assume that there exist feedback pathways with symmetric weights to feedforward pathways then the algorithm is local. Similarly, it is assumed that there paired error neurons, which is biologically questionable.

2) Speed of convergence
The reviewers noted that this model requires many more iterations to converge on the correct errors, and questioned the utility of a model that involves this much additional computational overhead.

The authors included some new text regarding biological plausibility and speed of convergence. They also included some new results to address some of the other concerns. However, there is still a core concern about the importance of this work relative to the original Whittington & Bogacz (2017) paper. It is nice to see those original results extended to arbitrary graphs, but is that enough of a major contribution for acceptance at ICLR? Given that there are still major issues related to (1) in the model, it is not clear that this extension to arbitrary graphs is a major contribution for neuroscience. And, given the issues related to (2) above, it is not clear that this contribution is important for ML. Altogether, given these considerations, and the high bar for acceptance at ICLR, a "reject" decision was recommended. However, the AC notes that this was a borderline case.